# Identification of a small molecule inhibitor that stalls splicing at an early step of spliceosome activation

Anzhalika Sidarovich[1], Cindy L Will[1]*, Maria M Anokhina[1], Javier Ceballos[2], Sonja Sievers[3], Dmitry E Agafonov[1], Timur Samatov[1†], Penghui Bao[1], Berthold Kastner[1], Henning Urlaub[4], Herbert Waldmann[2], Reinhard Lührmann[1]*

[1]Department of Cellular Biochemistry, Max Planck Institute for Biophysical Chemistry, Göttingen, Germany; [2]Department of Chemical Biology, Max Planck Institute of Molecular Physiology, Dortmund, Germany; [3]Compound Management and Screening Center, Max Planck Institute of Molecular Physiology, Dortmund, Germany; [4]Bioanalytics Group, Institute for Clinical Chemistry Göttingen, University Medical Center, Göttingen, Germany

**Abstract** Small molecule inhibitors of pre-mRNA splicing are important tools for identifying new spliceosome assembly intermediates, allowing a finer dissection of spliceosome dynamics and function. Here, we identified a small molecule that inhibits human pre-mRNA splicing at an intermediate stage during conversion of pre-catalytic spliceosomal B complexes into activated B$^{act}$ complexes. Characterization of the stalled complexes (designated B$^{028}$) revealed that U4/U6 snRNP proteins are released during activation before the U6 Lsm and B-specific proteins, and before recruitment and/or stable incorporation of Prp19/CDC5L complex and other B$^{act}$ complex proteins. The U2/U6 RNA network in B$^{028}$ complexes differs from that of the B$^{act}$ complex, consistent with the idea that the catalytic RNA core forms stepwise during the B to B$^{act}$ transition and is likely stabilized by the Prp19/CDC5L complex and related proteins. Taken together, our data provide new insights into the RNP rearrangements and extensive exchange of proteins that occurs during spliceosome activation.

*For correspondence: cwill1@mpibpc.mpg.de (CLW); reinhard.luehrmann@mpi-bpc.mpg.de (RL)

Present address: †Evotec International GmbH, Göttingen, Germany

Competing interests: The authors declare that no competing interests exist.

## Introduction

The precise excision of an intron from a pre-mRNA and the concomitant ligation of its flanking exons (pre-mRNA splicing) are catalysed by the spliceosome, a highly dynamic ribonucleoprotein (RNP) macromolecular complex. Spliceosomes assemble de novo on the pre-mRNA substrate in a stepwise manner by the sequential recruitment of five small nuclear ribonucleoprotein particles (snRNPs) and numerous non-snRNP factors (reviewed by *Will and Lührmann, 2006*; *Wahl et al., 2009*). Initially, the U1 snRNP binds to the 5' splice site (SS) and the U2 snRNP interacts stably with the branch site (BS) of the pre-mRNA, forming the A complex. Subsequently, association of the U4/U6.U5 tri-snRNP generates the B complex that remains catalytically inactive until it undergoes extensive compositional and conformational rearrangements, including the dissociation of U1 and U4, resulting in the formation of the B$^{act}$ complex. The latter is converted into a catalytically active spliceosome (designated B*) by the action of the RNA helicase Prp2 (*Kim and Lin, 1996*). The B* complex catalyses the first step of splicing, generating the cleaved 5' exon and intron-3' exon lariat intermediates, and at this stage the spliceosomal C complex is generated. After additional RNP rearrangements, the C complex catalyses the second step, resulting in the ligation of the 5' and 3' exons and release of the intron in the form of a lariat.

During spliceosome assembly and catalytic activation, the snRNA components of the snRNPs, together with the pre-mRNA, form a highly complex, dynamic RNA-RNA network (*Staley and Guthrie, 1998*). At the A complex stage, the U1 snRNA base pairs with the 5′SS of the pre-mRNA and the U2 snRNA basepairs with the BS. Upon integration of the U4/U6.U5 tri-snRNP, in which the U6 and U4 snRNAs are base paired and form two extended RNA helices (U4/U6 stem I and stem II) (*Agafonov et al., 2016*; *Nguyen et al., 2016*; *Wan et al., 2016b*), the 5′ end of U2 snRNA base pairs with the 3′ end of U6 snRNA, forming U2/U6 helix II (*Madhani and Guthrie, 1992*), and stem loop I of U5 snRNA interacts with exon nucleotides near the 5′SS (*Newman and Norman, 1992*; *Wyatt et al., 1992*; *Sontheimer and Steitz, 1993*). The U1 snRNA is displaced from the 5′SS of the pre-mRNA by the DEAD-box protein Prp28 (*Staley and Guthrie, 1999*), allowing the base pairing interaction between the 5′SS and the conserved ACAGA box sequence of the U6 snRNA (*Sawa and Abelson, 1992*; *Wassarman and Steitz, 1992*). A key step during the spliceosome activation phase is the unwinding of the U4/U6 helices by the RNA helicase Brr2 (*Laggerbauer et al., 1998*; *Raghunathan and Guthrie, 1998*). This enables U6 to form short duplexes with U2 (U2/U6 helix Ia and helix Ib), and a catalytically important U6 internal stem loop (that is, the U6 ISL), which together coordinate the metal ions important for splicing catalysis (*Fica et al., 2013*; *Hang et al., 2015*; *Galej et al., 2016*; *Yan et al., 2016*). The catalytic RNA-RNA network involving U2, U6, and U5 snRNA, that is ultimately established in the spliceosome is comprised of RNA structural elements very similar to the catalytic core of group II self-splicing introns (*Chan et al., 2012*; *Fica et al., 2014*; *Hang et al., 2015*; *Galej et al., 2016*; *Rauhut et al., 2016*; *Wan et al., 2016a*; *Yan et al., 2016*). After the first catalytic step of the splicing reaction, the spliceosome's catalytic RNA network must be further reorganized to align the step two reactants (*Smith and Konarska, 2008*; *Horowitz, 2012*).

The protein composition of the spliceosome is also highly dynamic, with extensive exchanges of proteins occurring from one step to another (*Fabrizio et al., 2009*; *Agafonov et al., 2011*). During the B to Bact transition, about 30 proteins leave the human spliceosome (*Bessonov et al., 2010*; *Agafonov et al., 2011*), including the U4/U6-specific and Lsm proteins, and also the B-specific proteins. The latter include hSnu23, RED, Smu1, MFAP, FBP21, hPrp38, NPW38 and NPW38BP, which bind during B complex formation and, like the U4/U6 and Lsm proteins, are missing or much less abundant in human Bact complexes (*Agafonov et al., 2011*). The function of most of the B-specific proteins is not clear, but they are not required for stable integration of the tri-snRNP during B complex formation (*Boesler et al., 2016*), and instead likely contribute to the activation process, as has been demonstrated for Prp38 (*Xie et al., 1998*; *Schütze et al., 2016*).

More than 35 proteins are recruited or stably-integrated during Bact formation. These include proteins belonging to the Prp19/CDC5L complex, which is thought to enter the spliceosome as a pre-assembled complex (*Makarova et al., 2004*; *Grote et al., 2010*). The yeast equivalent of the human Prp19/CDC5L complex - the NTC - has been shown to play an essential role in spliceosome activation, with NTC proteins stabilizing the interaction of U5 and U6 with the spliceosome after U4 has been released (*Chan et al., 2003*). Furthermore, NTC/NTC-related proteins are in intimate contact with the catalytically active RNA network within the Bact, C and post-catalytic ILS spliceosomes (*Yan et al., 2015*; *Galej et al., 2016*; *Rauhut et al., 2016*; *Wan et al., 2016a*; *Yan et al., 2016*). Proteins related to the Prp19/CDC5L complex, which are defined as such due to their physical or genetic interaction with components of this complex or their presence in the 35S U5 snRNP (*Makarov et al., 2002*), as well as several additional Bact proteins, are also recruited during activation. However, it is presently not clear whether all of the proteins exchanged at this stage are released/interact concomitantly or stepwise during the B to Bact transition, and thus whether there is a coordinated handover of one set of proteins with another. Given the extensive exchange of proteins during the transition from the B to Bact there are likely to be numerous additional, intermediate assembly stages in which only a subset of those proteins exchanged during these transitions are released or recruited. Information about the order of protein recruitment and release, and their temporal relationship to changes in the structure of the spliceosome's catalytic RNA, may shed light on the function of these proteins during activation.

Small molecules have proven highly useful to study many cellular processes such as transcription and translation, and their potential for dissecting the complex process of pre-mRNA splicing is becoming increasingly clear. Inhibitors of pre-mRNA splicing might enable the identification of novel, transient intermediates of the spliceosome, and thus allow a finer dissection of its dynamics

and function. Given the link between splicing and human diseases, including cancer (reviewed by *Padgett, 2012*; *Singh and Cooper, 2012*; *Scotti and Swanson, 2016*), small molecules that modulate the splicing process are also potentially of therapeutic value (*Bonnal et al., 2012*; *Ohe and Hagiwara, 2015*). Although the number of inhibitors of pre-mRNA splicing continues to grow, there remains a limited number of known compounds that stall pre-mRNA splicing at a defined stage. Several compounds with known cancer-inhibitory activity have been shown to inhibit splicing. These include, among others: (i) Spliceostatin A, Pladionelide B, Sudemycin, GEX1A and related compounds that target the U2-associated SF3b protein complex (*Kaida et al., 2007*; *Kotake et al., 2007*; *Fan et al., 2011*; *Hasegawa et al., 2011*) and stall splicing during the A complex stage (*Roybal and Jurica, 2010*; *Corrionero et al., 2011*; *Folco et al., 2011*), (ii) isoginkgetin (*O'Brien et al., 2008*) and (iii) 1,4 napthoquinones and 1,4-heterocyclic quinones, which block specifically the second catalytic step of splicing (*Berg et al., 2012*). Several other splicing inhibitors have been reported (*Patil et al., 2012*; *Samatov et al., 2012*; *Effenberger et al., 2013*; *Pawellek et al., 2014*; *Effenberger et al., 2015*), including inhibitors of protein acetylation/deacetylation (*Kuhn et al., 2009*) and of DNA topoisomerase I (*Tazi et al., 2005*). Finally, compounds that inhibit the kinase activity of members of the Clk/Sty, SRPK, or DYRK protein families, which phosphorylate serine-arginine-rich SR splicing factors and other spliceosomal proteins, were also identified and shown to modulate alternative splicing patterns in vivo (reviewed by *Ohe and Hagiwara, 2015*).

Here, we identify a small molecule (cp028) that inhibits pre-mRNA splicing at an intermediate stage of the spliceosome activation process. Spliceosomal complexes stalled by this compound (designated $B^{028}$) can be chased into catalytically active complexes by supplementing them with MN-digested nuclear extract, demonstrating that $B^{028}$ is not a dead-end complex, but rather a functional spliceosome intermediate. A detailed characterization of the $B^{028}$ complexes reveals that the large exchange of proteins that occurs during the conversion of the spliceosomal B complex to the activated $B^{act}$ complex proceeds stepwise, with the release of U4/U6 proteins occurring prior to and independently of the exchange of other spliceosomal proteins. RNA structure probing suggests that the U2/U6 RNA network in the $B^{028}$ complex differs from that of the spliceosomal $B^{act}$ complex, consistent with a role for Prp19/CDC5L complex proteins in generating the catalytic RNA network that is ultimately established in $B^{act}$. Our studies underscore the power of small molecule inhibitors of pre-mRNA splicing to study splicing mechanisms and to dissect the highly complex spliceosome activation process.

## Results

### Identification of a novel small molecule that inhibits pre-mRNA splicing in vitro

To identify novel compounds that inhibit pre-mRNA splicing in vitro, we utilized a previously described high throughput screening assay developed in our lab (*Samatov et al., 2012*). This luminescence-based assay measures the interaction of the DEAD-box protein DDX41 (Abstrakt) with immobilized pre-mRNA upon incubation under splicing conditions with HeLa nuclear extract. As DDX41 binds first when catalytically active spliceosomes are formed (*Bessonov et al., 2010*; *Agafonov et al., 2011*), compounds that inhibit spliceosome assembly during or prior to the C complex stage can be identified. Using an automated setup, we screened ~170,000 compounds at a concentration of 50 μM for splicing inhibition activity. 30 reproducibly-positive compounds were further tested in standard in vitro splicing assays. Eight compounds were confirmed to inhibit the splicing of $^{32}$P-labelled MINX pre-mRNA in HeLa nuclear extract. Here we have focused on one of them, denoted compound 028 (cp028) 1-(2-Ethylphenyl)−5-((5-(4-fluorophenyl)furan-2-yl) methylene) pyrimidine-2,4,6 (1H,3H,5H)−trione (*Figure 1A*).

In the presence of 50 μM cp028, splicing of MINX pre-mRNA in HeLa nuclear extract was reduced relative to the DMSO control (that is, the cp028 solvent) after 60 min, with splicing completely abolished at concentrations above 150 μM (*Figure 1B*). A fine titration of cp028 revealed an $IC_{50}$ value (where mRNA formation is inhibited by 50%) of $54 \pm 4$ μM (*Figure 1—figure supplement 1*). At inhibitory cp028 concentrations, an accumulation of spliceosomal complexes that run similar to A and B were observed (*Figure 1C*). A kinetic analysis of splicing complex formation revealed that in the presence of cp028, the amount of these complexes peaks already after 20 min (*Figure 1—figure*

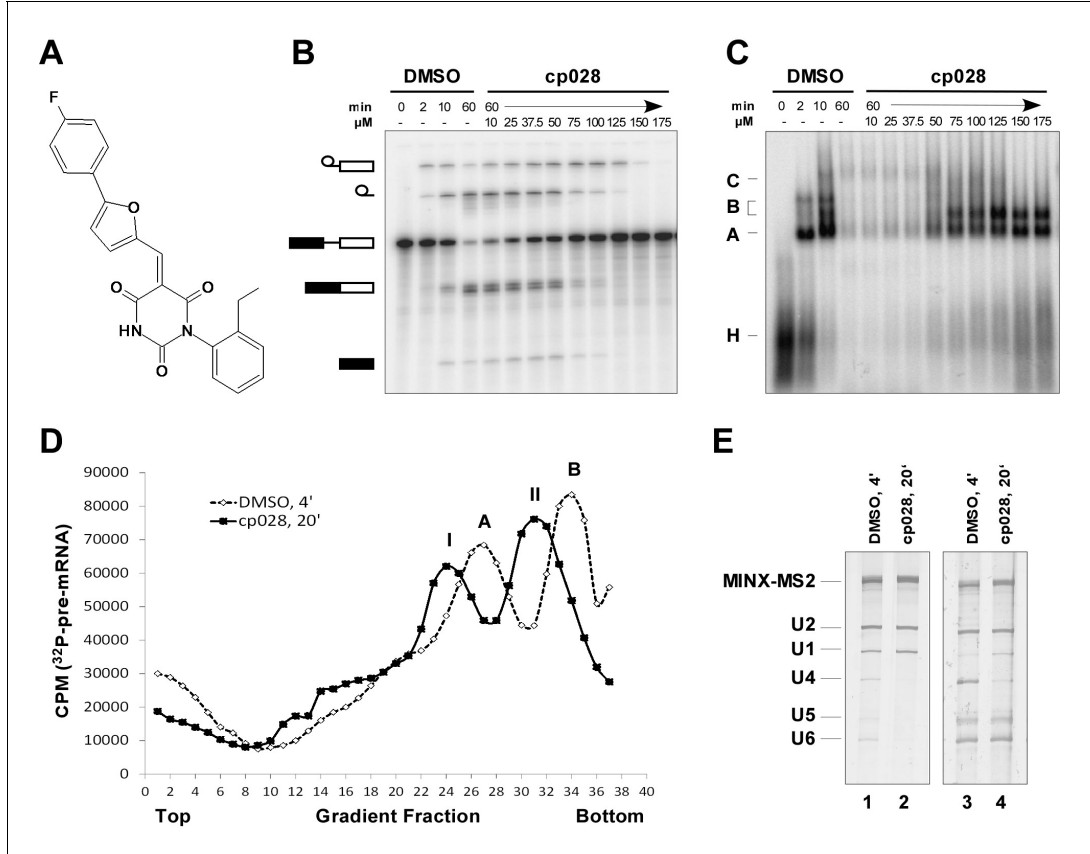

**Figure 1.** Characterization of spliceosomes that accummulate in the presence of the small molecule inhibitor cp028. (A) Structure of compound 028 (cp028). (B, C) Splicing was performed with $^{32}$P-MINX pre-mRNA in the presence of 10–175 μM compound 028 in HeLa nuclear extract for 60 min. RNA was analysed by denaturing PAGE (B) and spliceosomal complex formation was analysed on an agarose gel (C). The positions of the pre-mRNA, and splicing intermediates and products, or of spliceosomal complexes C, B, A, H are indicated on the left. Bands were visualized by autoradiography. DMSO, control reaction with the solvent. (D) Glycerol gradient sedimentation profile of the spliceosomal complexes formed under splicing conditions for 20 min in the presence of cp028 or 4 min in the presence of solvent (DMSO). (E) RNA composition of splicing complexes formed in the presence of cp028 or DMSO that were affinity purified from peak A or I (lanes 1 and 2) and peak B or II (lanes 3 and 4) of the glycerol gradient. RNA was analysed by denaturing PAGE and detected by silver staining. RNA identities are indicated on the left.

The following figure supplements are available for figure 1:

**Figure supplement 1.** Determination of IC$_{50}$ and kinetics of splicing complex formation in the presence of compound 028.

**Figure supplement 2.** Identification of abundant proteins in affinity-purified A$^{028}$ complexes.

**Figure supplement 3.** Most U4/U6.U5 tri-snRNPs remain intact in the presence of cp028.

*supplement 1*). Thus, in higher eukaryotes cp028 inhibits the A to B complex transition, and also the transition of the B complex to later spliceosomal complexes. Similar results were obtained with an IgM pre-mRNA substrate (*Figure 1—figure supplement 1*), indicating a general inhibitory effect of cp028 on pre-mRNA splicing in vitro. Interestingly, cp028 did not inhibit splicing of actin pre-mRNA in *S. cerevisiae* whole cell extracts at a concentration up to 250 μM (*Figure 1—figure supplement 1*), suggesting that it targets or interferes with one or more spliceosomal component specifically involved in splicing in higher eukaryotes, and further that it does not inhibit splicing in a non-discriminatory manner.

## Characterization of spliceosomal A complexes stalled in the presence of compound 028

To analyse the stalled spliceosomal complexes in more detail, we purified them by MS2 affinity-selection after subjecting them to glycerol gradient centrifugation. Spliceosomes assembled in the presence of cp028 migrated in two main peaks (I and II) of the gradient, which were each shifted to slightly lower S-values compared to the corresponding main peaks of the 4 min DMSO control reaction, which contain A and B complexes, respectively (*Figure 1D*). Spliceosomes affinity-purified from peak I contained stoichiometric amounts of U1 and U2 snRNAs, plus the MINX-MS2 pre-mRNA (*Figure 1E*), indicating that they are spliceosomal A complexes. Mass spectrometry (MS) analyses of these complexes, designated A[028], revealed primarily U1 and U2 snRNP proteins, with only a few differences in other splicing factors compared to 'kinetic' A complexes, purified after splicing for 4 min (*Supplementary file 1*). To determine which proteins are present in stoichiometric or near stoichiometric amounts, we performed 2D gel electrophoresis followed by MS. Abundant proteins (with a molecular mass above 25 kDa) in the A[028] complex included essentially all U1 and U2 snRNP proteins, plus U2AF65, U2AF35, SRSF1, SRSF7, hnRNPA1, and SPF30 (*Figure 1—figure supplement 2*). A nearly identical set of abundant/moderately-abundant proteins is found in kinetically-stalled A complexes (*Agafonov et al., 2011*), indicating that A[028] is very similar to A complexes formed in the absence of inhibitor.

## Most U4/U6.U5 tri-snRNPs are intact in the presence of cp028

A block in the conversion of the A complex to the pre-catalytic B complex could arise via disruption of the U4/U5.U6 tri-snRNP. To check if the stability of the U4/U6.U5 tri-snRNP is affected by cp028, we incubated HeLa nuclear extract under splicing conditions (without addition of pre-mRNA) in the presence of DMSO or cp028, and then separated the spliceosomal snRNPs by glycerol gradient centrifugation and determined their distribution by Northern blotting. In the DMSO control, tri-snRNPs peaked in fractions 15 to 18, based on the presence of the U4, U5 and U6 snRNAs (*Figure 1—figure supplement 3*). In the presence of cp028, tri-snRNPs sedimented as a broader peak in fractions 15 to 21, suggesting that they may be more prone to aggregation (*Figure 1—figure supplement 3*), and only a small increase in the amount of free U5 and U4/U6 snRNPs was observed in the upper fractions of the gradient, indicating that most of the tri-snRNP remains intact. We also performed immunoprecipitations at different salt concentrations with antibodies directed against the U5-116K/Snu114 protein and determined the extent of coprecipitation of U4 and U6 snRNA (indicating tri-snRNP formation). In the presence of cp028 only a small to moderate decrease in the amount of U4/U6 snRNP that coprecipitated with U5 was observed at 190 and 290 mM salt, compared to the DMSO control (*Figure 1—figure supplement 3*). Thus, cp028 has only a moderate affect on tri-snRNP stability, with most tri-snRNPs still intact under splicing conditions.

## Compound 028 stalls spliceosome assembly at a stage between B and B[act]

Affinity-purified complexes formed in the presence of cp028, that were isolated from gradient peak II, contained stoichiometric amounts of U2, U5, U6 snRNA and the MINX-MS2 pre-mRNA, but substantially reduced levels of both U1 and U4 snRNAs (*Figure 1E*), suggesting that they are stalled at the B[act] stage after Brr2-mediated dissociation of U4 snRNA. However, human B[act] complexes exhibit a slightly higher S-value than B complexes during glycerol gradient centrifugation, and also migrate considerably slower on native agarose gels than B (*Bessonov et al., 2010*). The B-like complex that accumulates in the presence of cp028 (henceforth designated B[028]) has, in contrast, a lower S-value than B and also migrates faster than a B[act] complex on a native gel, suggesting that it is a novel assembly intermediate.

MS analysis (*Supplementary file 1*) and 2D gel electrophoresis followed by MS (*Figure 2A,B*) revealed that B[028] complexes, like B complexes, contain predominantly U2 and U5 snRNP proteins, as well as most proteins from the group designated B-specific (RED, MFAP1, hSmu-1, hPrp38, hSnu23, NPW38 and NPW38BP) that are recruited at the B complex stage and are no longer abundant/present in the mature human B[act] complex (*Bessonov et al., 2010*; *Agafonov et al., 2011*). Abundant proteins of purified B[028] complexes, as determined by 2D gel electrophoresis, are summarized in *Figure 2B*. U4/U6 proteins, such as Prp3, Prp4 and Prp31, are present in very low amounts

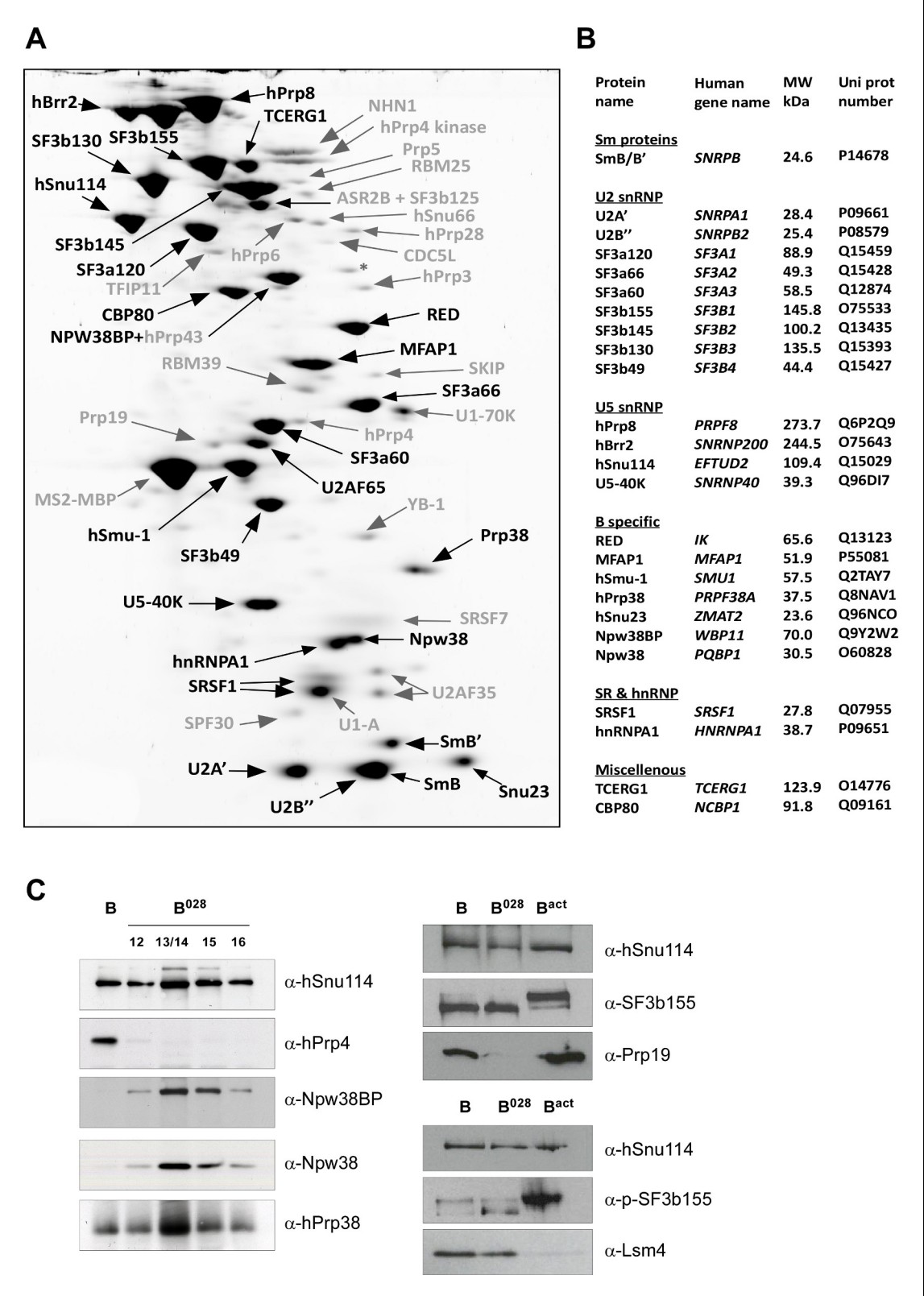

**Figure 2.** Identification of abundant proteins in affinity-purified B[028] complexes. (**A**) Proteins (larger than 25 kDa) from MS2 affinity-purified B[028] complexes assembled on MINX-MS2 pre-mRNA were separated by 2D gel electrophoresis, stained with RuBPS, and the identities of single protein spots were determined by mass spectrometry. Abundant proteins were identified by visual inspection and are labelled black, and less abundant ones grey. (**B**) Summary of abundant proteins detected in B[028] complexes. Proteins are grouped according to their association with snRNPs or stage of

*Figure 2 continued on next page*

*Figure 2 continued*

recruitment. (C) Proteins from affinity-purified B, B[028] or B[act] complexes (as indicated) were analysed by Western blotting using antibodies against the indicated proteins. Antibodies against hSnu114 were used to ensure equal loading. Left panel: Western blots with proteins from the B[028] peak fractions 12–16 (as indicated above each lane) of a glycerol gradient on which affinity-purified B[028] complexes were fractionated.

in B[028], consistent with loss of the U4 snRNA (*Supplementary file 1*, *Figure 2*). In addition, only minor amounts of the tri-snRNP-specific proteins hSnu66 (110K) and hSAD1 (65K) are detected in B[028]. In contrast, the U6-associated Lsm 2–8 proteins, which like the U4/U6 and tri-snRNP-specific proteins are normally lost during the conversion of B to B[act], are still abundant based on the total spectral counts of peptides sequenced by MS (*Supplementary file 1*) and also on immunoblotting with antibodies against the Lsm4 protein (*Figure 2C*). B[act]-specific proteins, which are only abundant in the B[act] complex but not in C, and Prp19/CDC5L complex and related proteins, as well as other B[act] proteins, which are recruited or more stably associated in B[act] complexes (*Bessonov et al., 2010*; *Agafonov et al., 2011*), are absent or highly underrepresented in B[028]. The U2 snRNP SF3b155 protein is hyperphosphorylated during catalytic activation of the spliceosome (*Wang et al., 1998*; *Girard et al., 2012*). Immunoblotting with anti-SF3b155 antibodies that recognize both non-phosphorylated SF3b155 and its slower-migrating, hyperphosphorylated form, as well as with an antibody specific for phosphorylated SF3b155, demonstrated that SF3b155 is not hyperphosphory-lated in B[028] complexes (*Figure 2C*). Taken together, these results indicate that B[028] complexes are stalled at an intermediate stage between B and B[act], after displacement of U4 snRNA by Brr2 and loss of the U4/U6-specific proteins, but before release of the B-specific and Lsm proteins, and recruitment/stable integration of B[act]-specific and Prp19/CDC5L proteins. Assuming that the B[028] complex does not represent an atypical assembly intermediate, these data indicate that the large exchange of proteins during the B to B[act] transition occurs stepwise, with the release of U4/U6 proteins occurring prior to and independently of the exchange other spliceosomal proteins (see Discussion).

## Characterization of the RNA-RNA network in B[028] complexes via psoralen crosslinking

During activation, the RNA-RNA network of the spliceosome is substantially rearranged. To elucidate the nature of the spliceosomal RNA-RNA network in the B[028] complex, we first performed UV-induced psoralen crosslinking. For comparative purposes, we also analysed purified B and B[act] complexes in parallel. To purify sufficient amounts of the latter complexes, it was necessary to use a truncated pre-mRNA substrate lacking a 3' exon and containing a shortened polypyrimidine tract (that is, PM5-10) (*Figure 3—figure supplement 1*). B[028] complexes formed on PM5-10 pre-mRNA have a nearly identical RNA and protein composition compared to those formed on MINX pre-mRNA (*Figure 3—figure supplement 1* and data not shown). RNA-RNA crosslinks were analysed via Northern blotting by sequentially incubating with [32]P-labeled probes against U1, U2, U4, and U6 snRNAs, and also by autoradiography of the [32]P-labeled PM5-10 pre-mRNA on which the B, B[028] and B[act] complexes were assembled. (*Figure 3—figure supplement 1*). U4/U6 and U4/U6/U2 cross-links were observed with B, but not B[028] and B[act], consistent with the loss of the vast majority of U4 in the latter two complexes, while a U1/pre-mRNA crosslink was observed in both B and B[028] complexes due to the presence of small amounts of U1 in these complexes. A U2/U6 crosslink, which based on its migration behaviour is U2/U6 helix II (*Anokhina et al., 2013*), was detected in all complexes, albeit less efficiently in B and B[028], indicating base pairing between the 5' and 3' end of U2 and U6, respectively. The reduced efficiency of the U2/U6 crosslink in B and B[028] might be due to the presence of the Lsm proteins, which are located in the human tri-snRNP near the region of U6 that forms U2/U6 helix II and thereby may hinder crosslinking (*Agafonov et al., 2016*). Likewise, a U6/pre-mRNA crosslink, consistent with a U6 ACAGA box/5'SS base pairing interaction, was also detected in all three complexes, but less efficiently in B[028] (*Figure 3—figure supplement 1*).

## RNA structure probing indicates that U6 snRNA is rearranged in the B[028] complex

To determine the structure of the RNA network in the B[028] complex, the Watson/Crick (W/C) edges of RNA were probed with 1-cyclohexyl-3-(2-morpholinoethyl) carbodiimide metho-*p*-toluene sulphonate (CMCT, U-specific with low reactivity towards G), dimethylsulfate (DMS, A- and C-specific), or *β*-ethoxy-*α*-ketobutyraldehyde (kethoxal, G-specific). Reactivity with these reagents is observed solely with unpaired nucleotides, providing information about RNA secondary structure. After incubating purified B[028] and B[act] complexes with CMCT, DMS or kethoxal, RNA was isolated and primer extension assays were performed with 5'-end labeled oligonucleotides complementary to chosen RNA sequences; if chemical modification takes place, elongation of the primer by reverse transcriptase (RT) will be stopped one nucleotide before the chemically modified RNA nucleotide. Accessibility to a given chemical was designated strong, medium or weak depending on the intensity of the reverse transcriptase stop (minus the background observed without DMS, CMCT or kethoxal), which was quantitated using a PhosphorImager (*Figure 3—figure supplement 2*).

Consistent with our psoralen crosslinking results, U2/U6 helix II appears to be formed in all complexes, based on the lack of chemical modification of U2 nts 3–11 and U6 nts 88–95 (which form helix II), as evidenced by the absence of RT stops above background at these nucleotides (*Figure 3* and *Figure 3—figure supplement 2*). Chemical modification also supported the formation of the base pairing interaction between the U6 ACAGA box (nts A40-A44) and intron nucleotides near the 5'SS in B[028]. That is, U6 nts A40-A44, as well as PM5-10 intron nts + 4 to +8 (relative to the 5'SS) were protected from modification in purified B[028] and B[act] complexes (*Figure 3* and *Figure 3—figure supplement 3*). U2 nucleotides that base pair with the BPS (G33-A38), with the exception of U2-G33, were also not modified in B[028] or B[act], consistent with the formation of the U2/BPS helix. In yeast, U2 snRNA toggles between different intra-molecular conformations, forming U2 helix IIc instead of SLIIa (*Hilliker et al., 2007*; *Perriman and Ares, 2007*). However, human U2 nts A88-A95 and U53-U60, which would be base paired if helix IIc were formed, are accessible in B[028] and B[act] complexes, suggesting that U2 helix IIc is not formed in B[028] complexes, nor in B[act], as previously shown (*Anokhina et al., 2013*) (*Figure 3—figure supplement 4*).

Previous work from our lab showed that the catalytically important U2/U6 snRNA interaction forms a three helix (three-way) junction in B[act] complexes (*Anokhina et al., 2013*) (*Figure 3*). However, an alternative U2/U6 conformation—that is, a four helix (four-way) junction, was also previously proposed (*Sun and Manley, 1995*; *Sashital et al., 2004*) (*Figure 3*). In the latter, the U6 ISL forms an extended lower stem, which precludes formation of U2/U6 helix Ib, and U2 instead forms an internal stem-loop (SL1). Most nucleotides comprising the U6 ISL loop (G67-A70) are accessible in B[act], whereas nucleotides forming the stem (U57-U64 and A70-A77) are not accessible or only very weakly accessible (with the exception of A70 and G75), consistent with the formation of the ISL (*Figure 3* and *Figure 3—figure supplement 2*). U6 ISL loop nucleotides (A69, C68, C66) are also accessible in B[028], but to a lower extent compared to those in B[act]; this enhanced protection of ISL loop nucleotides in B[028] could potentially be due to shielding by bound proteins. Nucleotides of the ISL stem are highly protected in B[028], suggesting that the ISL stem is indeed formed. Interestingly, U6 nts 79–81 are protected in B[028], but not in B[act], suggesting that the lower part of the ISL stem is extended in B[028] (*Figure 3*), consistent with the formation of a four-way U2-U6 junction (*Figure 3*), in contrast to the three-way junction that forms in B[act]. The base pairing partners of U6-G80 and U6-C81 in the four-way junction model—that is, U6-C55 and U6-G54 - are also protected in B[028]; however, this would also be the case if U2/U6 helix Ib were formed, as is observed with B[act]. The U2 chemical modification pattern in this region is consistent with the formation of both U2/U6 helix Ib (three-way) and the U2 SL1 (four-way), given that the observed, moderate modification of its loop-closing G-C base pair (C14 and G19) is attributed to breathing of this short U2 helix (*Figure 3* and *Figure 3—figure supplement 2*). U6 nts A50-U52 involved in U2/U6 helix Ia formation are relatively well-protected in both B[028] and B[act], with only weak modification of A50, while G49 is accessible in both B[act] and B[028] (*Figure 3* and *Figure 3—figure supplement 2*). In addition, U2 nts involved in helix Ia formation (G25-C28) are also protected against modification (*Figure 3* and *Figure 3—figure supplement 2*), consistent with the formation of U2/U6 helix Ia in both the B[028] and B[act] complex. Taken together, our results indicate that the U6 ISL and U2/U6 helix Ia are formed in B[028], but that the conformation of the U2/U6 junction in B[028] differs from that in B[act], possibly forming a four-way

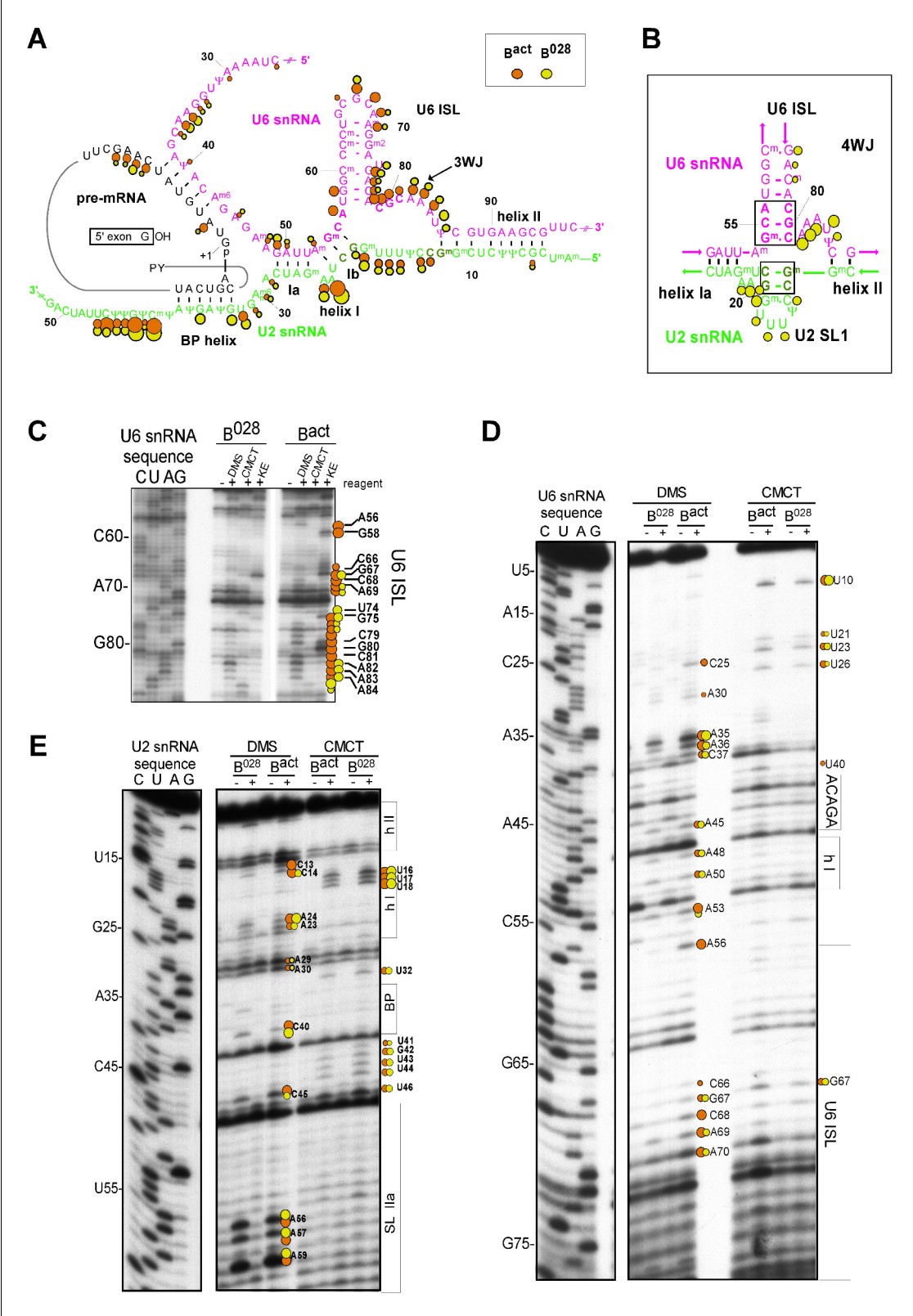

**Figure 3.** Characterization of the catalytic U2/U6 RNA network in the B[028] complex. (A) Summary of the chemical modification patterns of U6 and U2 snRNA in B[028] and B[act] complexes. The U2/U6 interaction network is shown forming a 3-way junction (3WJ). The different size of the dots indicates different degrees of chemical modification. (B) 4-way junction model of U2/U6 snRNAs with summary of U2 and U6 chemical modification pattern in B[028] complexes. (C, D) Primer extension analysis of U6 from CMCT, DMS and KE modified B[028] and B[act] complexes. Primer extension was performed

*Figure 3 continued on next page*

*Figure 3 continued*

with an oligonucleotide complementary to U6 nts 80–100 (**C**) or to the additional nucleotides that were added to the 3' end of the U6 snRNA (**D**). (**E**) Primer extension analysis of U2 from CMCT and DMS B$^{028}$ and B$^{act}$ complexes, performed with an oligonucleotide complementary to U2 nts 75–95.

The following figure supplements are available for figure 3:

**Figure supplement 1.** Identification of RNA-RNA interactions in the B$^{028}$ complex via psoralen crosslinking.

**Figure supplement 2.** Structure probing of the U2 and U6 snRNAs in purified B$^{028}$ complexes with kethoxal.

**Figure supplement 3.** RNA structure probing of the pre-mRNA in B$^{028}$ and B$^{act}$ complexes.

**Figure supplement 4.** U2 snRNA intramolecular helix IIc is not formed in the B$^{028}$ complex.

as opposed to a three-way junction. This, in turn, is consistent with the idea that the catalytic RNA network of the spliceosome is rearranged in multiple steps during catalytic activation.

## B$^{028}$ complexes can be chased into catalytically-active spliceosomes

To determine whether the B$^{028}$ complex is a functional assembly intermediate, as opposed to a dead end complex, we performed chase experiments. When affinity-purified B$^{028}$ complexes were incubated under splicing conditions in the absence of extract, no splicing was observed (*Figure 4*). However, when micrococcal nuclease (MN) treated nuclear extract—in which all endogenous snRNPs were degraded—was added, both catalytic steps of splicing were observed, at levels similar to that observed with purified B complexes plus MN-treated extract (*Figure 4*). Splicing was not observed when pre-mRNA alone was incubated under splicing conditions with MN-treated extract (*Figure 4*). Thus, purified B$^{028}$ can be chased into a mature spliceosome that catalyzes both steps of splicing, demonstrating that it is a functional intermediate. However, when cp028 was added to the MN-treated extract, no splicing was observed after incubation with both B and B$^{028}$ complexes (*Figure 4*). This suggests that cp028 binds to and/or inactivates a splicing factor required for the conversion of the B/B$^{028}$ complexes into catalytically active spliceosomes.

## The B$^{028}$ complex shares structural features with the B complex, but has a distinct morphology

The B$^{028}$ complex appears to be stalled at an intermediate stage between the spliceosomal B and B$^{act}$ complex. To elucidate its structure, we analysed affinity-purified B$^{028}$ complexes by negative-stain electron microscopy (EM) after gradient fixation (GraFix). An overview of the negatively-stained raw images revealed a homogeneous population of monodisperse particles of the same basic morphology with a maximum length of ~40 nm (*Figure 5A*). The majority of B$^{028}$ particles exhibit a roughly triangular shape, with a lower foot-like protrusion. Classification and class averaging of these single particle images (*Figure 5B*) revealed that in the most dominant classes (comprising ca. 40%) the B$^{028}$ complex exhibits a triangular shape with a long density element on the left and an upper more globular domain (*Figure 5B*, columns 1–2). Classes with a mushroom-like shape (columns 3–4, rows 1–3) were observed much less frequently (in ca. 20% of the class averages), as were classes showing both mushroom and triangular shape features (columns 3–4, last row; ca. 20%). The basic morphology of the various EM views of the B$^{028}$ complex indicates that it shares similarities with the spliceosomal B complex, when the triangular views are considered, and similarities with the B$^{act}$ complex, when the mushroom views are compared. That is, the human B complex exhibits a triangular or rhombic shape, with a foot region and elongated density element along one side (the so-called foot-stump axis), similar to B$^{028}$. The upper end of this main axis appears somewhat different in both complexes. In addition, the head domain of the B complex points more upward compared to B$^{028}$, generating a more rhombic shape in the vast majority of classes, whereas the globular (head) domain in B$^{028}$ is orientated more toward the central region of the particle, leading to its isosceles triangle shape. Thus, despite many similarities, there appear to be clear structural differences between B$^{028}$ and B complexes. Although in some cases B$^{028}$ exhibited a mushroom-like shape typical for the

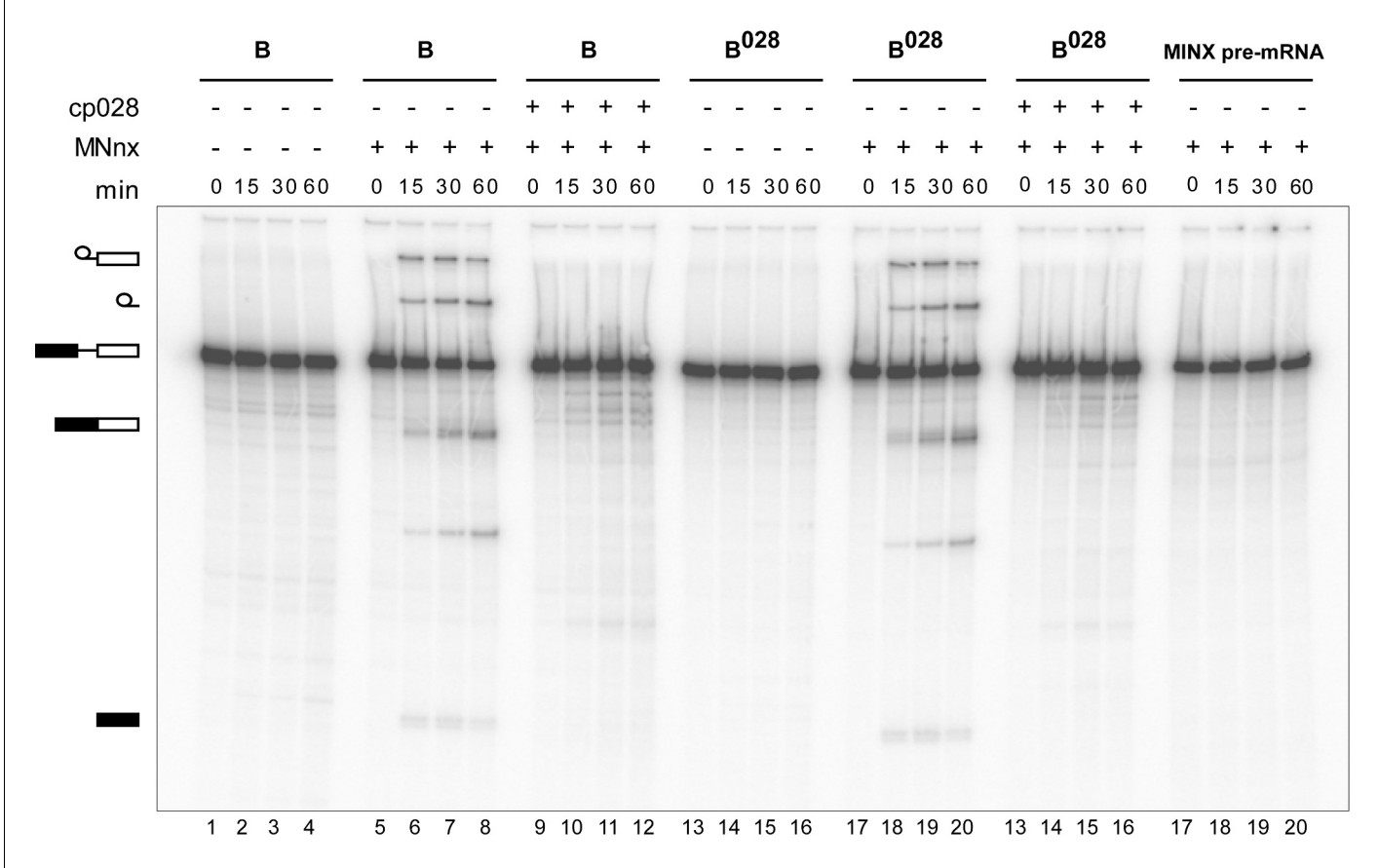

**Figure 4.** Purified B[028] complexes can be chased into catalytically-active spliceosomes. Affinity-purified B or B[028] complexes (as indicated above) formed on MINX-MS2 pre-mRNA, were incubated at 30°C for the indicated times (0–60 min) under splicing conditions in the presence of buffer alone or micrococcal nuclease treated HeLa nuclear extract (MNnx) lacking or containing 150 µM cp028. RNA was analysed by denaturing PAGE and visualized with a Phosphorimager. The positions of the pre-mRNA, splicing intermediates and products are indicated on the left.

human B[act] complex, structural similarities between B[act] and B[028] are limited in the predominant views of the latter, and structural details seen in B[028] are not recognizable in B[act]. This may in part be due to differences in their adsorption to the carbon film, which would lead to different views of B[028] versus B[act].

## Structure activity relationship correlation

To identify chemical features that determine the splicing inhibition activity of cp028, we purchased or synthesized a variety of cp028 analogues and then tested their effect on pre-mRNA splicing in vitro (*Figure 6*, *Figure 6—figure supplements 1*, *2* and *Figure 6—source data 1*). Whereas most of the structural variations of the *p*-fluorophenyl group had little or no effect on activity (*Figure 6*, *Figure 6—figure supplements 1* and *2*), introducing fluorine or chlorine at the meta-position of the fluorophenyl group (analogues 343867 and 343871) compared to fluorine at the para-position (as found in cp028), enhanced slightly the splicing inhibition effect (*Figure 6*, *Figure 6—figure supplements 1* and *2*). Replacing the fluorine of the *p*-fluorophenyl group with a phenyl group (analogue 343868) or replacing the *p*-fluorophenyl group by a piperidine (analogue 344353) (*Figure 6*, *Figure 6—figure supplements 1* and *2*) abolished inhibition activity completely. In addition, removing the furan group also reduced the potency dramatically (analogues 343878 and 343879). Thus, the fluorophenyl group and the furan group are required for inhibiting pre-mRNA splicing in vitro and likely play an important role in target recognition. Nearly all modifications in the ethylphenyl group (*Figure 6*, *Figure 6—figure supplements 1* and *2*), or its omission (analogue 343135), had only

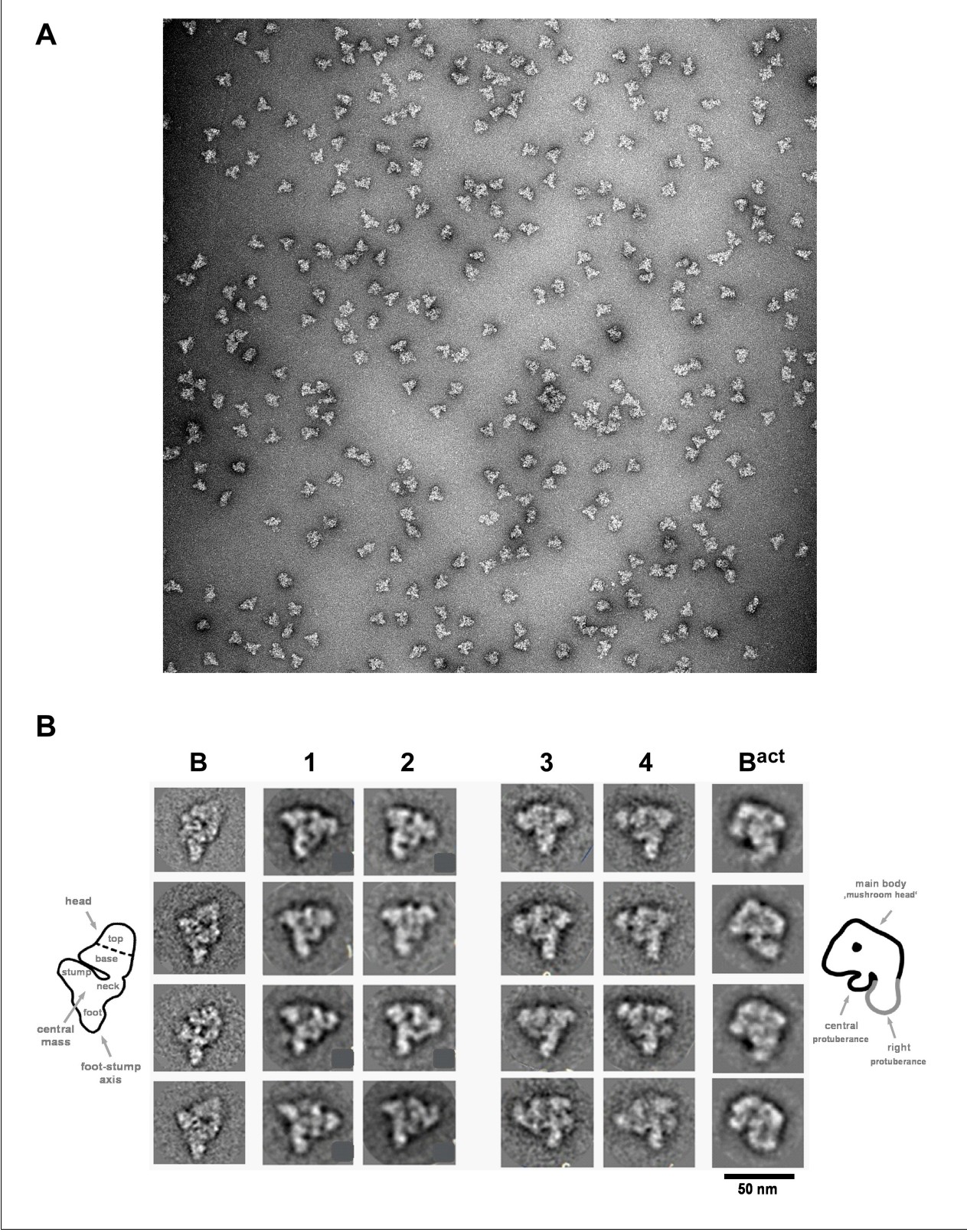

**Figure 5.** Morphology of the B028 complex under the electron microscope. (**A**) Overview of raw images of affinity-purified B028 complexes obtained by negative stain electron microscopy. (**B**) Selected class averages (columns 1–4) of the B028 complex, compared to published EM structures of human B and Bact complexes. Schematic representations of the B complex and Bact complex are shown at the far left or right, respectively, with main structural

*Figure 5 continued on next page*

*Figure 5 continued*

features labeled. Scale bar corresponds to 50 nm. B complexes with a triangular shape similar to that shown typically represent more than 80% of the EM images whereas, approximately 60% of purified B$^{act}$ complexes exhibit a mushroom-like shape.

minimal or no substantial effect on the potency of cp028, suggesting that this structural feature it is not essential for splicing inhibition in vitro. Thus, this site could potentially be used to introduce a tag to identify the target(s) of compound 028 in the splicing reaction. Notably, reduction of the con-jugated double bond (analogue 343139, *Figure 6C*) completely abolished splicing inhibition,

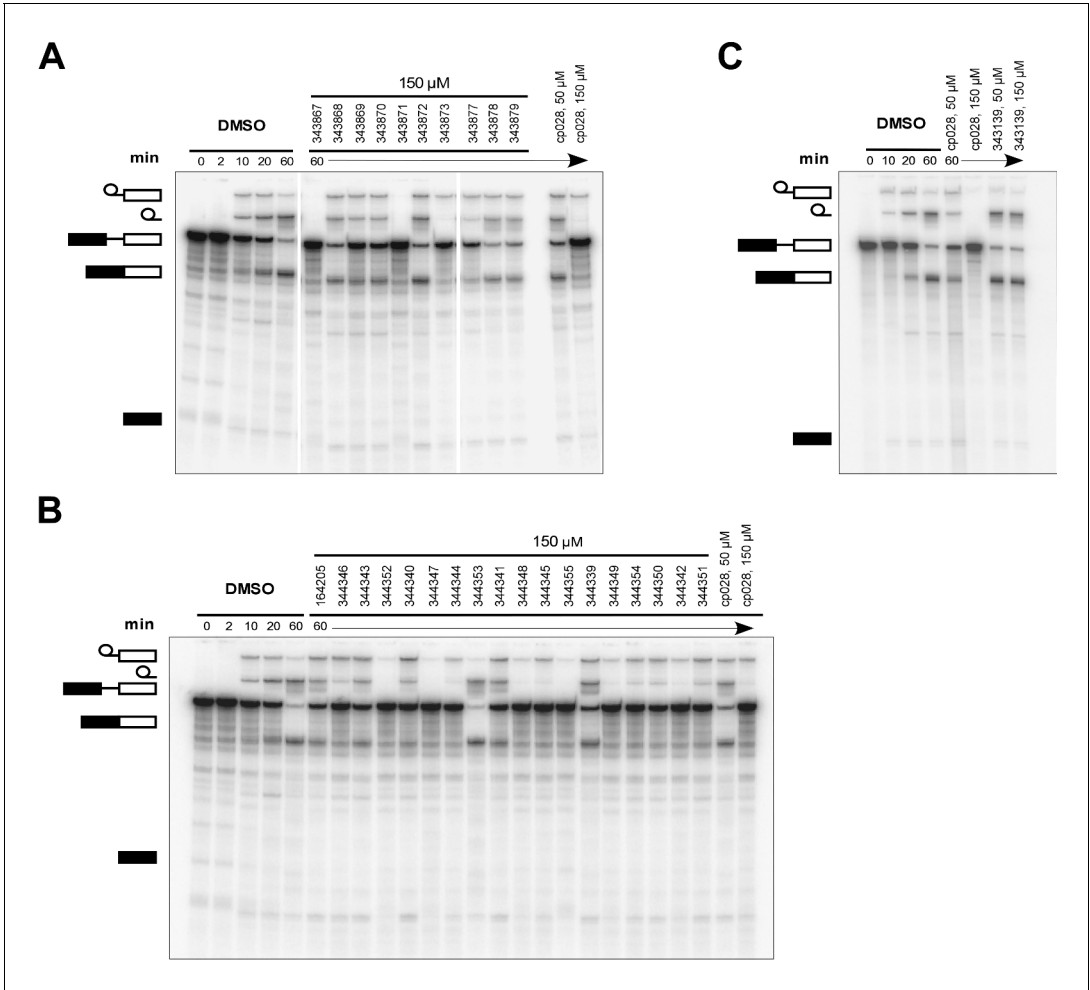

**Figure 6.** Effects of alterations in the structure of cp028 on its splicing inhibition activity. (**A–C**) In vitro splicing was performed with $^{32}$P-labelled MINX pre-mRNA and HeLa nuclear extract for 60 min after addition of cp028 and various analogues of cp028 at the indicated concentrations. The structures of the tested analogues are summarized in *Figure 6—figure supplement 1*. RNA was analysed by denaturing PAGE and the positions of the pre-mRNA, and splicing intermediates and products are indicated on the left.

The following source data and figure supplements are available for figure 6:

**Source data 1.** NMR spectra of cp028 analogues synthesized in house.

**Figure supplement 1.** Structures of cp028 analogues and their effect on pre-mRNA splicing in vitro.

**Figure supplement 2.** Effect of cp028 analogues on in vitro splicing.

suggesting that cp028 likely exerts its inhibitory activity by covalently modifying one or several target proteins and/or that the rotational freedom afforded by a single bond may be deleterious for its inhibition activity.

## Discussion

### A new tool to dissect pre-mRNA splicing at the spliceosome activation stage

We performed a high throughput screen, previously developed in our lab, followed by standard in vitro splicing assays in HeLa nuclear extract, to identify small molecule inhibitors of mammalian splicing. One of the confirmed hit compounds, termed cp028, inhibited splicing in HeLa nuclear extract with an $IC_{50}$ value of 54 μM. Cp028 not only hindered spliceosome assembly at the A complex stage, but also led to the accumulation of a novel spliceosome assembly intermediate ($B^{028}$), stalled during the extensive rearrangements of the B complex that ultimately yield the $B^{act}$ complex. Importantly, purified $B^{028}$ complexes could be chased into catalytically active complexes by supplementing them with MN-digested nuclear extract, indicating that $B^{028}$ is not a dead-end complex, but rather a functional spliceosome intermediate. However, we cannot rule out that $B^{028}$ might represent an atypical intermediate that is not normally formed during the spliceosome activation phase, but that nonetheless can be chased into an active spliceosome. Cp028 stalls the spliceosome assembly process at a novel stage during activation, distinguishing it from previously reported small molecule inhibitors of splicing that block spliceosome assembly at earlier stages (*O'Brien et al., 2008*; *Kuhn et al., 2009*; *Roybal and Jurica, 2010*; *Corrionero et al., 2011*; *Folco et al., 2011*) or first during the catalytic steps of splicing (*Berg et al., 2012*). Thus, cp028 represents a new tool to study pre-mRNA splicing mechanisms, in particular to dissect/modulate the highly complex spliceosome activation process.

### The exchange of proteins during the B to $B^{act}$ transition occurs stepwise

Our knowledge about the spliceosome's highly complex compositional dynamics is hampered by the limited number of distinct spliceosomal complexes that can be biochemically detected or isolated. The progression from a precatalytic B to $B^{act}$ spliceosomal complex involves multiple RNP rearrangements, and the recruitment and release of a large number of proteins in both higher and lower eukaryotes. During this step the U4 snRNA, U4/U6 proteins, U6-associated Lsm proteins and so-called B-specific proteins are destabilized/released from the spliceosome, whereas $B^{act}$-specific proteins, components of the Prp19/CDC5L complex, and a group of proteins designated Prp19/CDC5L-related are recruited. Whether these remodelling events occur concomitantly or in a stepwise fashion, and the order of protein requirement and release has remained unclear. Our MS and 2D gel analyses of purified $B^{028}$ complexes, together with western blotting, suggest that the loss of the U4/U6 proteins (hPrp3, hPrp4, hPrp31, 20K and 15.5K), presumably during the Brr2-mediated unwinding of the U4/U6 duplex, can occur independently of the release of the U6 Lsm proteins or B-specific proteins. Likewise, the recruitment of the human Prp19/CDC5L complex and related proteins, as well as $B^{act}$-specific proteins, appear to occur after the release of the U4/U6-specific proteins. However, the order of these subsequent compositional changes remains unclear. A number of studies including very recent single molecule studies in yeast, indicate that the recruitment of the NTC occurs after release of the U4 snRNA (*Tarn et al., 1993*; *Chan et al., 2003*; *Hoskins et al., 2016*), which is consistent with U4/U6 protein release preceeding recruitment of the vast majority of the Prp19/CDC5L complex. Release of the U6 snRNA from U4 allows rearrangements in the structure of the U6 snRNA. The formation of a rearranged U6 structure, including formation of the U6 ISL, may provide new binding sites that enable the recruitment/stable binding of the Prp19/CDC5L complex and related proteins, and/or $B^{act}$ proteins. In yeast, the NTC was reported to be required for the release of the Lsm proteins from the 3'-end of U6 snRNA (*Chan et al., 2003*), thus the interaction or stable association of the Prp19/CDC5L complex and related proteins with the spliceosome may be a prerequisite for the subsequent loss of the Lsm proteins from the spliceosome during activation. Indeed, recent cryo-EM studies of the *S. cerevisiae* spliceosomal $B^{act}$ complex, revealed that several NTC proteins (e.g. Syf1, Syf2 and Syf3) are close to U2/U6 helix II and thus to the 3' end of the U6 snRNA (*Rauhut et al., 2016*; *Yan et al., 2016*). As the Lsm proteins bind the 3' end of U6

(*Achsel et al., 1999*), it is likely that their binding and that of the aforementioned NTC proteins is mutually exclusive.

## A potential role for the Prp19/CDC5 complex and related proteins in establishing the U2/U6 RNA three-way junction in B$^{act}$ complexes

In B$^{028}$, the U4/U6 duplex has been unwound and U4 snRNA, as well as the U4/U6 snRNP proteins, have been destabilized/released from the spliceosome. This frees U6 snRNA to engage in new intra- and inter-molecular base pairing interactions, leading to the catalytically active RNA network within the spliceosome. Our structure probing data (*Figure 3*) suggest that in B$^{028}$ the U6 snRNA has rearranged and formed the functionally important ISL, but that the lower stem of the U6-ISL is extended by three base pairs. This would prevent formation of U2/U6 helix Ib and is consistent with the formation of a U2/U6 four-way (four RNA helix) junction. As U2 and U6 have been shown to form a three-way junction in human B$^{act}$ complexes (*Figure 3*)(*Anokhina et al., 2013*), the apparently different structure of U2/U6 in B$^{028}$ may represent an intermediate conformation that forms directly after release of U6 from the U4 duplex by Brr2. This would be consistent with the idea that the catalytic RNA conformation forms stepwise during the B to B$^{act}$ transition.

In yeast, U2 and U6 have been shown to form a three-way junction (*Madhani and Guthrie, 1992*; *Hilliker et al., 2007*; *Burke et al., 2012*), but there is also evidence that a competing four-way junction can form, at least in vitro (*Sashital et al., 2004*; *Guo et al., 2009*). U2/U6 four-helix junction formation had been documented previously only with protein-free snRNAs (*Sashital et al., 2004*; *Guo et al., 2009*), and it is thus likely that spliceosomal proteins promote the formation of a U2/U6 three-helix junction. Subsequent conversion of the apparent U2/U6 four-way junction observed in B$^{028}$, to a three-way junction, may require the interaction of proteins normally recruited/stabilized during activation, and/or the loss of the Lsm or B-specific proteins, that are normally released prior to B$^{act}$ formation. In the yeast B$^{act}$ complex, the catalytic U2/U6 RNA network is contacted not only by the U5 Prp8 protein, but also by Cef1 (CDC5L in human) and Prp46 (Prl1 in human), which are present in the human Prp19/CDC5L complex, and the Prp19/CDC5L-related proteins Syf3, Prp45 (SKIP/SNW1 in human) and Cwc2 (RBM22 in human) (*Rasche et al., 2012*; *Rauhut et al., 2016*; *Yan et al., 2016*). Given the multiple contacts that the Prp19/CDC5L complex and related proteins make with the catalytic RNA network of the spliceosome already at the B$^{act}$ stage, it is very likely that they act as chaperones that help to form/maintain the active conformation of the RNA network. Indeed, genetic experiments in *S. cerevisiae* indicate that Cwc2 helps to stabilize U2/U6 helix 1 (*Hogg et al., 2014*). As the U2/U6 three and four helix conformations are competing structures, and B$^{028}$ complexes lack the Prp19/CDC5L complex and related proteins, these proteins may shift the equilibrium toward the three helix structure once they are stably incorporated into the spliceosome.

## Potential mechanisms whereby cp028 stalls spliceosome assembly and activation

While the target(s) of cp028 remain unclear, the fact that two different steps of the spliceosome assembly pathway are stalled, suggest that cp028 may have multiple spliceosome-related targets. Cp028 did not inhibit splicing in *S. cerevisiae* whole cell extracts, at concentrations where splicing in HeLa nuclear extract was completely inhibited. This suggests that it targets or interferes with one or more spliceosomal components (or potentially the posttranslational modification thereof) that is/are specifically involved in splicing in higher eukaryotes. Indeed, while there is conservation of the core spliceosomal components between yeast and man, mammalian spliceosomes contain a large number of proteins not found in *S. cerevisiae* (*Fabrizio et al., 2009*).

The ability of a large percent of A complexes to be converted in the presence of cp028 to B complexes that undergo the initial steps of activation and can be chased into catalytically-active spliceosomal complexes, indicates that a substantial portion of the tri-snRNP is not severely affected by cp028. Indeed gradient centrifugation and co-immunoprecipitation experiments indicated that most tri-snRNPs are intact under splicing conditions. The conversion of the spliceosomal A complex into a B complex with stably associated U4/U6.U5 tri-snRNP is dependent not only on an intact tri-snRNP, but also on several splicing factors, including SPF30 (*Rappsilber et al., 2001*; *Meister et al., 2001*), the U4/U6.U5 tri-snRNP proteins 65K/hSAD1 and 110K/hSART1 (*Makarova et al., 2001*), SR proteins (*Roscigno and Garcia-Blanco, 1995*), the kinases hPrp4K (*Schneider et al., 2010*) and SRPK2

(*Mathew et al., 2008*) and the DEAD-box helicase Prp28 (*Staley and Guthrie, 1999*; *Mathew et al., 2008*; *Boesler et al., 2016*). Thus, one or more of these proteins are potentially targeted directly or indirectly by cp028. It is also conceivable that cp028 affects the structure or composition of the A complex such that stable tri-snRNP integration is impaired. However, the protein composition of affinity-purified $A^{028}$ complexes was nearly identical to that of A complexes formed in the absence of cp028, except $A^{028}$ no longer contained stoichiometric amounts of the DEAD box protein Prp5 and the splicing factor SPF45, both of which are not involved in B complex formation (*Ruby et al., 1993*; *Lallena et al., 2002*; *Liang and Cheng, 2015*).

The mechanism whereby $B^{028}$ stalls the spliceosome activation process is also currently unclear. Release of the Lsm proteins and B-specific proteins is blocked, as well as recruitment/stable association of $B^{act}$ proteins and Prp19/CDC5L complex and related proteins. Thus, one or more of these proteins may be targetted by cp028, or alternatively the binding partners of these proteins in the B complex may be affected/compromised. Our structure-activity relationship studies revealed chemical groups that do not contribute to the inhibition activity of cp028, such as the primary ethyl group. Thus, modified versions of cp028 containing a tag at this position can be generated and potentially used in future experiments to identify its target(s) in the spliceosome.

## Materials and methods

### High throughput screening of small molecules

Approximately 170,000 compounds from the compound library of the Compound Management and Screening (COMAS) Center of the Max Planck Society were screened using a semi-automated setup as described below (see also *Samatov et al. (2012)*). The COMAS library mainly consists of commercial screening compounds, which were chosen according to diversity and drug-likeness, but also contains natural-product-like substances and known drugs. Prior to screening, the amount of antibody, washing volumes and the concentration of the nuclear extract were optimized to yield a screening-ready assay with an assay window of approximately 9 and a Z' value of ca. 0.5. For high throughput screening, black 384 well Greiner Fluotrac 600 high-binding microplates (Cat. No. 781077) were first coated with rabbit anti-MBP antibodies in 20 mM HEPES-KOH (pH 7.9) (0.02 mg/ml, 10 μl/well) by incubating at 27°C overnight with shaking. The plates were subsequently washed twice using 50 μl bottom wash and 50 μl well washing options (Elx405 Washer, Biotek, VT, USA) with crosswise final aspiration of washing buffer (1 mM HEPES-KOH, pH 7.9, 0.075 mM $MgCl_2$, 7.5 mM NaCl, 0.005% Tween20). Next, 50 nl of compound (10 mM in DMSO) were added by acoustic dispensing technology (Echo520, Labcyte Inc., Sunnyvale, Ca, USA). Directly after that, 10 μl of a splicing mixture containing 40% (v/v) HeLa nuclear extract (containing Flag-tagged Abstract protein), 0.1 μg/ml anti-FLAG-HRP antibodies, 0.1 pmoles of PM5 pre-mRNA (*Bessonov et al., 2008*) bound by MS2-MBP, 24 mM HEPES-KOH (pH 7.9), 65 mM KCl, 2 mM ATP, 20 mM creatine phosphate, 3.0 mM $MgCl_2$, and 0.1% Tween 20, were dispensed into the assay plates (Multidrop Combi, Thermo Fisher Scientific, MA, USA). PM5 pre-mRNA containing MS2 aptamers was pre-incubated with MS2-MBP for 30 min on ice (in 100 mM Hepes-KOH, pH 7.9). The plates were shaken briefly using a Tele-shake (Inheco, Germany) and the splicing mixtures were incubated for 2 hr at 25°C. Plates were subsequently washed twice with washing buffer using 300 μl bottom wash and 50 μL well wash options of the Elx405 Washer with crosswise final aspiration. Subsequently, 10 μl of SuperSignal ELISA Femto Substrate (Pierce, cat. no. 37075) were dispensed into the assay plates (Multidrop Fisher Combi, Thermo Scientific, MA, USA) and the luminescence signal was read after 8 min incubation using a Spectramax Paradigm reader (Molecular Devices, Sunnyvale, Ca, USA). Positive and negative controls were included on every plate. The positive control contained all assay components, except 50 nl DMSO alone was added, and in the negative control the PM5 pre-mRNA was omitted. Primary screening identified ca. 2700 compounds that reduced the luminescence signal to less than 50% compared to the DMSO-treated positive control. These primary hit compounds were retested in duplicate using the same assay conditions. Thirty of these reduced the luminescence signal to less than 50% of the DMSO control in both replicates and were thus further analyzed in standard in vitro splicing reactions using the MINX pre-mRNA substrate (*Zillmann et al., 1988*), leading to the identification of eight small molecule inhibitors of splicing (that is, a false positive rate of ca 1.6%).

## Pre-mRNA splicing and splicing complex formation

HeLa S3 cells were obtained from GBF, Braunschweig (currently Helmholtz Zentrum für Infektionsforschung, Braunschweig) and tested negative for mycoplasma. HeLa nuclear extract was prepared essentially as previously described (*Dignam et al., 1983*). In vitro splicing reactions contained 40% (v/v) HeLa nuclear extract, 65 mM KCl, 3 mM MgCl$_2$, 2 mM ATP, 20 mM creatine phosphate, 0.2 mM DTE and 10 nM uniformly $^{32}$P-labelled, m$^7$G-capped MINX pre-mRNA or IgM pre-mRNA (*Guth et al., 1999*), plus DMSO (maximally 5%) or compound 028 1-(2-Ethylphenyl)—5-((5-(4-fluorophenyl)furan-2-yl)methylene) pyrimidine-2,4,6 (1H,3H,5H)-trione (synthesized in-house or purchased from ChemDiv, cat. no. 3679-1153) dissolved in DMSO, and were incubated at 30°C for the indicated times. RNA was recovered and analysed on a 14% denaturing polyacrylamide gel. Unspliced pre-mRNA, and splicing intermediates and products were detected using a Typhoon phosphoimager (GE Healthcare) or by autoradiography. The formation of spliceosomal complexes was analysed by agarose gel electrophoresis in the presence of 0.4 µg/µl heparin (*Das and Reed, 1999*). For yeast in vitro splicing, *S. cerevisiae* cell extracts from the yeast strain BJ2168 were prepared as previously described (*Fabrizio et al., 2009*). Splicing with yeast whole cell extract was performed with $^{32}$P-labelled MS2 actin pre-mRNA at 23°C for 20 min in 10 µl volume, containing 60 mM KPO$_4$ (pH 7.4), 3% PEG-8000, 2.5 mM MgCl$_2$, 0.2 mM DTT, 2 mM spemidine and 2 mM ATP, plus DMSO or cp028. Reactions were then subjected to proteinase K digestion with 50 mM EDTA and 1% SDS. RNAs were recovered by PCl extraction and ethanol precipitation, and analysed on an 8% polyacrylamide/8M urea gel.

## MS2 affinity purification of spliceosomal complexes

In vitro assembled spliceosomal complexes were purified by gradient centrifugation, followed by MS2 affinity-selection using amylose beads (NEB) essentially as previously described (*Bessonov et al., 2008*). Briefly, $^{32}$P-labelled MINX or PM5-10 pre-mRNA containing MS2 aptamers at its 3' or 5' end, respectively, was incubated with a 20-fold molar excess of MS2-MBP protein. Standard splicing reactions containing 10 nM pre-mRNA were incubated at 30°C for 4 min (A complex), 4 min (B complex), 20 min (A$^{028}$ and B$^{028}$ complexes) or 170 min (B$^{act}$ complex). To remove contaminating B complexes, a 25-fold molar excess of DNA oligonucleotides (M6 and M12 oligonucleotides) complementary to the PM5-10 5' exon were subsequently added to the B$^{act}$ reaction, which was then incubated at 30°C for an additional 20 min as previously described (*Bessonov et al., 2010*). The splicing reaction was separated on a linear 15–30% (v/v) glycerol gradient containing G150 buffer (20 mM Hepes-KOH, pH 7.9, 150 mM KCl, 1.5 mM MgCl$_2$) by centrifugation for 18 hr at 30000 rpm (or 28000 rpm for B$^{act}$ complexes) in a Sorvall TST 41.14 rotor. Peak fractions were loaded onto amylose beads (NEB), and after washing with G150 buffer, bound spliceosomal complexes were eluted with 20 mM maltose in G150 buffer. Note that kinetically-stalled A and B complexes also contain low amounts of other subsequently formed complexes that migrate on the gradient near A or B, respectively. RNA was recovered from the purified complexes, separated on a denaturing polyacrylamide gel and visualized by silver staining.

## 2D gel electrophoresis, western blotting and mass spectrometry

Two-dimensional gel electrophoresis of A$^{028}$ and B$^{028}$ complexes, and the subsequent identification of individual protein spots via mass spectrometry were performed essentially as described previously (*Agafonov et al., 2011*), except an 8% acrylamide gel was used for the second dimension and proteins were stained with Coomassie. For western blotting, proteins of affinity-purified complexes were separated by SDS-PAGE and transferred to a nitrocellulose membrane (Protran, Whatman). The membrane was incubated with antibodies against the following human proteins: hSnu114 (*Fabrizio et al., 1997*), SF3b155 (*Will et al., 2001*) phospho-SF3b155 (*Girard et al., 2012*), Lsm4 (*Achsel et al., 1999*), hPrp4/60K (*Makarova et al., 2004*), hPrp19 (*Makarova et al., 2004*), hPrp38, Npw38 and Npw38BP. Bound antibody was detected using an ECL detection kit (GE Healthcare). Proteins associated with affinity-purified spliceosomal complexes were identified by LC-MSMS as described previously (*Boesler et al., 2015*).

## Immunoprecipitation and glycerol gradient centrifugation of snRNPs

HeLa nuclear extract that was pre-incubated under splicing conditions for 20 min with DMSO or 150 µM cp028, was fractionated on a 10–30% (v/v) glycerol gradient in G150 buffer by centrifugation in a Sorval TST41.14 rotor at 27,000 rpm for 18 hr at 4°C. Gradients were fractionated manually from the top. Alternatively, the extract was incubated at 4°C with end-over-end rotation for 4 hr with Protein A Sepharose (PAS) pre-bound by anti-hSnu114 antibodies. The PAS with bound material was washed with IPP buffer (10 mM Tris-HCl, pH 8.0, 0.1% v/v NP-40) containing 190, 290 or 390 mM NaCl, as indicated. RNA was recovered from the gradient fractions or PAS by proteinase K digestion, followed by phenol/chloroform extraction and ethanol precipitation, and identified by Northern blotting with $^{32}$P-labelled probes against U1, U2, U4, U5, and U6 snRNA.

## Psoralen crosslinking

Affinity-purified spliceosomal complexes (0.5–1.0 pmole) formed on $^{32}$P-labelled PM5-10 pre-mRNA (*Bessonov et al., 2010*) in G150 buffer were supplemented with 40 µg/ml of 4'-aminomethyl-4,5',8-trimethylpsoralen hydrochloride (AMT). After incubating for 10 min on ice, samples (+/- AMT) were irradiated with 365 nm UV-light for 30 min at 4°C with a distance of 4 cm between the samples and UV lamp. RNA was resolved on a 5% denaturing polyacrylamide gel and transferred to a nylon membrane (Hybond XL, GE Healthcare). The membrane was hybridized sequentially with $^{32}$P-labelled probes against the U1, U2, U4, and U6 snRNAs. Prior to incubation with a different probe, $^{32}$P-labelled probe was removed by boiling the membrane for 30 min in 15 mM NaCl, 1.5 mM sodium citrate and 0.1% (w/v) SDS. Efficient probe removal was controlled using a Typhoon phosphoimager.

## Chemical modification of RNA and primer extension analyses

Chemical modification of MS2-affinity purified B, B$^{028}$ and B$^{act}$ complexes (assembled on PM5-10 pre-mRNA) was performed essentially as described previously (*Anokhina et al., 2013*). To analyse the extreme 3' end of U6, 3'-extended U6 molecules were generated as previously described (*Anokhina et al., 2013*). For primer extension analysis, oligonucleotides complementary to a given region of an snRNA or the pre-mRNA were $^{32}$P-labelled at their 5' end using T4 polynucleotide kinase. Primer extension with reverse transcriptase was performed as described previously (*Hartmuth et al., 1999*). Primer extension products were analysed on denaturing 9.6% polyacrylamide - 8.3 M urea sequencing gels and visualized by autoradiography. Data were quantified on a Typhoon 8600 phosphorimager as previously described (*Anokhina et al., 2013*) using Quantity ONE (BioRad). Signals were divided into three classes according to their fold increase over background.

## Chase of B$^{028}$ complexes with micrococcal nuclease-treated extract

HeLa nuclear extract was treated with micrococcal nuclease (MN) as described previously (*Makarov et al., 2002*). Affinity-purified B or B$^{028}$ complexes formed on $^{32}$P-labelled MINX-MS2 pre-mRNA were incubated with splicing buffer alone (65 mM KCl, 3 mM MgCl$_2$, 2 mM ATP, 20 mM creatine phosphate, 20 mM Hepes-KOH, pH 7.9), or additionally in the presence of 20% (v/v) MN-treated HeLa nuclear extract containing 0 or 150 µM cp028. A 10-fold excess of unlabeled MINX-MS2 pre-mRNA was added to prevent the reassembly of snRNPs (that potentially dissociate from the purified complexes) on the radiolabeled pre-mRNA. The reaction was incubated at 30°C for 0 to 60 min. RNA was recovered, separated on a 14% denaturing polyacrylamide gel, and visualized with a Typhoon phosphoimager.

## Electron microscopy

For electron microscopy, affinity-purified complexes were subjected to a second, linear 10–30% (v/v) glycerol gradient containing G-150 buffer, and 0–0.1% (v/v) glutaraldehyde (*Kastner et al., 2008*). The samples were centrifuged at 50000 rpm for 4.75 hr in a Sorvall TH660 rotor at 4°C and 175 µl fractions were collected from the bottom with a fraction collector. Particles in peak fractions were negatively stained by the single-carbon film method (*Rigo et al., 2015*). Images were recorded at 160 kV and a magnification of 88000x with a CM200 FEG electron microscope (Philips, Netherlands) at RT on a 4 k x 4 k CCD camera (TVIPS, Germany). 3094 individual single-particle B$^{028}$ images were collected and subjected to single particle image-processing using the software package IMAGIC-5 (*van Heel et al., 1996*). After a reference-free alignment, images were subjected to multivariate

statistical analysis and classification (*Dube et al., 1993*; *van Heel and Frank, 1981*; *van Heel, 1984*). The resulting class averages were used as reference images in subsequent rounds of alignment until the class averages were stable.

## Synthesis of cp028 analogues

All reactions involving air- or moisture-sensitive reagents or intermediates were carried out in flame-dried glassware under an argon atmosphere. Dry solvents (THF, toluene, MeOH, DMF) were used as commercially available. Analytical thin-layer chromatography (TLC) was performed on Merck silica gel aluminium plates. Compounds were visualized by irradiation with UV light or potassium permanganate staining. Column chromatography was performed using silica gel Merck 60 (particle size 0.040–0.063 mm). $^1$H-NMR and $^{13}$C-NMR were recorded on a Bruker DRX400 (400 MHz), Bruker DRX500 (500 MHz) and INOVA500 (500 MHz) at 300 K using CDCl3 or (CD3)2SO as solvents (see *Figure 6—source data 1* for NMR spectra of the synthesized compounds). All resonances are reported relative to TMS. Spectra were calibrated relative to the solvent's residual proton and carbon chemical shift: CDCl3 ($\delta$ = 7.26 ppm for 1 hr NMR and $\delta$ = 77.16 ppm for 13C NMR); (CD3)2SO: $\delta$ = 2.50 ppm for 1 hr NMR and $\delta$ = 39.52 ppm for 13C NMR). Multiplicities are indicated as: br s (broadened singlet), s (singlet), d (doublet), t (triplet), q (quartet), quin (quintet), m (multiplet), and coupling constants (J) are given in Hertz (Hz). High resolution mass spectra were recorded on a LTQ Orbitrap mass spectrometer coupled to an Acceka HPLC-System (HPLC column: Hypersyl GOLD, 50 mm × 1 mm, particle size 1.9 µm, ionization method: electron spray ionization). Preparative HPLC separations were carried out using a reversed-phase C18 column (RP C18, flow 20.0 mL/min, solvent A: 0.1% TFA in water, solvent B: 0.1% TFA in acetonitrile, from 10% B to 100% B). All other chemicals and solvents were purchased from Sigma-Aldrich, Fluka, TCI, Acros Organics, ABCR and Alfa Aesar. Unless otherwise noted, all commercially available compounds were used as received without further purifications. Compounds 344355 (8015–3146), 344353 (3679–1405), 344342 (3679–1141), 344345 (3679–1145), 344347 (3679–1147), 344350 (3679–1155), 344349 (3679–1154), 344346 (3679–1146), 344348 (3679–1151), 344352 (3679–1170), 344344 (3679–1143), 344354 (3657–2730), 344351 (3679–1168), 344343 (3679–1143), 344339 (2277–0005), 164502 (0327–0135), 344340 (0327–0183) and 344341 (0327–0199) were purchased from ChemDiv (catalog number in parentheses) and their purity was determined by HPLC-MS prior to use. Compounds: cp028, 343869, 343870, 343877, 343873, 343867, 343871, 343878, 343879, 3434872, 343135 and 343868 were synthesized with the general strategy shown in *Scheme 1*. Starting materials 2-ethylaniline and 5-bromo-2-carbaldehyde (5-bromofurfural), and the different boronic acids were purchased from Aldric or Alfa Aesar. Urea **B** was obtained according to a procedure reported previously (*Laudien and Mitzner, 2001*). $^1$H NMR spectra of the compounds were in accordance with literature data. **D** compounds were obtained by Suzuki coupling employing conditions reported previously (*Martín-Matute et al., 2003*). Final compounds were synthesized from compound **B** in two steps as described (*Hussein et al., 2013*). All **cp028** analogues present were readily dissolved in CHCl$_3$ but had only low solubility in other organic solvents. All analogues were obtained as E/Z isomeric mixtures.

**Scheme 1.** General Synthesis of cp028 (R = 4fluoro) and analogues.

## 1-(2-Ethylphenyl)pyrimidine-2,4,6 (1H,3H,5H)-trione (C)

Sodium (0.21g, 9.14 mmol) was placed in an oven-dried flask under argon flow. Dry ethanol (50 mL, 0.18M) was cannulated into the flask and the mixture was stirred until all sodium had dissolved. Diethyl malonate (1.39 mL, 9.14 mmol) is then added, the mixture was stirred for 5 min and finally

urea **B** (1.50 g, 9.14 mmol) was added. The reaction was stirred under reflux for 16 hr. Solvents were evaporated. The crude residue was redissolved in NaOH aq. (2 M) and the mixture was washed with EtOAc to remove the remaining starting urea. The pH of the aqueous phase was adjusted to pH 1.0 with HCl aq. (1 M) leading to precipitation of the product. Product **C** (1.13 g, 53%) was isolated as a white solid pure enough to continue the synthesis. The product can be recrystallized from a mixture of EtOH/water (1:1) for further purification. [1]H NMR(DMSO, 400MHz): δ 1.07 (t, J = 7.6 Hz, 3 hr), 2.43 (q, J = 7.8 Hz, 2 hr), 3.67 (d, J = 20.9 Hz, 1 hr), 3.91 (d, J = 20.9 Hz, 1 hr), 7.16 (d, J = 7.7 Hz, 1 hr), 7.24–7.28 (m, 1 hr), 7.35 (m, 2 hr), 11.50 (s, 1 hr).

**Chemical structure 1.** 1-(2-Ethylphenyl)pyrimidine-2,4,6 (1H,3H,5H)-trione (C).

## 5-(4-Fluorophenyl)furan-2-carbaldehyde (D₁)

5-Bromofurfural (1.75g, 10.00 mmol), *p*-fluoroboronic acid (1.40 g, 10 mmol), potassium carbonate (3.46 g, 25 mmol), benzyltrimethylammoniun bromide (2.3 g, 10 mmol) and palladium acetate (0.05 g, 0.2 mmol) were suspended in water (14 mL, 0.71 M). The reaction was allowed to stir overnight at room temperature. The aqueous mixture was extracted with EtOAc and solvent was evaporated. After purification by flash chromatography (20% EtOAc/P.E) compound **D₁** was isolated as a white solid (1.86 g, 98%). [1]H NMR (CDCl₃, 400MHz): δ 6.78 (d, J = 3.7 Hz, 1 hr), 7.14 (t, J = 8.7 Hz, 2 hr), 7.31 (d, J = 3.7 Hz, 1 hr), 7.79–7.84 (m, 2 hr), 9.65 (s, 1 hr), in agreement with the literature (*Zhang et al., 2016*).

**Chemical structure 2.** 5-(4-Fluorophenyl)furan-2-carbaldehyde (D₁).

**Scheme 2.** Aldehydes used for the synthesis of the cp028 analogues.

All **D** aldehydes used are reported in the literature and most of them are commercially available. However, here all **D** aldehydes were synthesized following the procedure for compound **D₁**, and their identity was confirmed by 1H NMR (1H NMR spectra are given for those analogues where a [1]H NMR spectrum was not reported in the literature, that is, **D₂,₃,₆,₉**). Compound **D₈** (*Trinh et al.,*

2014). Compound **D$_7$** (*Mitsch et al., 2004*). Compound **D$_{10}$** (*Liégault et al., 2009*). Compound **D$_4$** (*Cosner et al., 2009*). Compound **D$_5$** (*Murasawa et al., 2012*). Compound **D$_{11}$** (*Umezawa et al., 2009*).

Compound **D$_2$** (95%), **$^1$H NMR (CDCl$_3$, 400MHz):** δ 3.85 (s, 6 hr), 6.51 (t, *J* = 2.3 Hz, 1 hr), 6.82 (d, *J* = 3.7 Hz, 1 hr), 6.96 (d, *J* = 2.3 Hz, 2 hr), 7.31 (d, *J* = 3.7 Hz, 1 hr), 9.65 (s, 1 hr). Compound **D$_3$** (87%), **$^1$H NMR (CDCl$_3$, 400MHz):** δ 6.86 (d, *J* = 3.7 Hz, 1 hr), 7.32 (d, *J* = 3.7 Hz, 1 hr), 7.36–7.41 (m, 2 hr), 7.69–7.71 (m, 1 hr), 9.68 (s, 1 hr). Compound **D$_6$** (80%), **$^1$H NMR (CDCl$_3$, 400MHz):** δ 6.86 (d, *J* = 3.7 Hz, 1 hr), 7.09 (t, *J* = 8.3, 2.5 Hz, 1 hr), 7.32 (d, *J* = 3.7 Hz, 1 hr), 7.42 (td, *J* = 8.0, 5.8 Hz, 1 hr), 7.52 (dt, *J* = 9.6, 2.3 Hz, 1 hr), 7.60 (ddd, *J* = 7.8, 1.5, 0.9 Hz, 1 hr), 9.68 (s, 1 hr). Compound **D$_9$** (74%), **$^1$H NMR (CDCl$_3$, 400MHz):** δ 6.88 (d, *J* = 3.7 Hz, 1 hr), 7.34 (d, *J* = 3.7 Hz, 1 hr), 7.38 (t, *J* = 7.3 Hz, 1 hr), 7.47 (t, *J* = 7.5 Hz, 2 hr), 7.63 (d, *J* = 7.4 Hz, 1 hr), 7.69 (d, *J* = 8.7 Hz, 2 hr), 7.90 (d, *J* = 8.7 Hz, 2 hr).

## (E/Z)−1-(2-Ethylphenyl)−5-((5-(4-fluorophenyl)furan-2-yl)methylene) pyrimidine-2,4,6 (1H,3H,5H)-trione (1:1 mixture of E/Z isomers) (cp028)

Aldehyde **D$_1$** (0.050 g, 0.263 mmol) was added to a solution of barbituric acid **C** (0.069 g, 0.263 mmol) in EtOH (3.3 mL, 0.08 M), leading to the appearance of an intense orange color. The reaction was heated at reflux for 2 hr and the solvents were evaporated. The crude product was purified by flash chromatography (1% MeOH/DCM) to obtain product **cp028** as an orange solid (83.0 mg, 78%). The product was isolated as a 1:1 mixture of the isomers E and Z. **$^1$H NMR (CDCl$_3$, 400MHz):** δ 1.13 (t, *J* = 7.6 Hz, 3 hr), 2.44 (q, *J* = 7.6 Hz, 2 hr), 6.95 (dd, *J* = 4.0, 0.6 Hz, 1 hr), 7.08–7.14 (m, 3 hr), 7.25–7.30 (m, 1 hr), 7.33–7.39 (m, 2 hr), 7.79 (dd, *J* = 9.0, 5.2 Hz, 2 hr), 7.90 (s, 1 hr), 8.43 (s, 1 hr), 8.74 (d, *J* = 3.9 Hz, 1 hr). **$^{13}$C NMR (CDCl$_3$, 100MHz):** δ 14.0, 24.1, 109.4, 111.4, 116.6, 116.7, 125.0, 125.1, 127.2, 128.0, 128.8, 129.4, 129.9, 132.7, 133.1, 140.9, 141.6, 149.7, 150.9, 161.0, 161.9, 163.3, 163.4, 165.0. **$^{19}$F NMR (CDCl$_3$, 100MHz):** δ −108.1 ppm. **HRMS:** calc. for [M+H]$^+$ C$_{23}$H$_{18}$N$_2$O$_4$F: 405.12451, found 405.12437

**Chemical structure 3.** (E/Z)−1-(2-Ethylphenyl)−5-((5-(4-fluorophenyl)furan-2-yl)methylene)pyrimidine-2,4,6 (1H,3H,5H)-trione (1:1 mixture of E/Z isomers) (cp028).

## (+/-)-1-(2-Ethylphenyl)−5-((5-(4-fluorophenyl)furan-2-yl)methyl) pyrimidine-2,4,6 (1H,3H,5H)-trione (racemic mixture) (343139)

To a solution of **cp028** (0.050 g, 0.124 mmol) in EtOH (1.2 mL, 0.1 M) at 0°, sodium borohydride (0.014 g, 0.371 mmol) was added. The mixture was allowed to reach room temperature and stirred for 1 hr before evaporating the solvents. To the crude product, 1M HCl was added and a white solid precipitated, which was filtered and purified by preparative HPLC to yield compound **343139** (32.8 mg, 65%) as a white solid. **$^1$H NMR (CDCl$_3$, 400MHz):** δ 0.75 (t, *J* = 7.6 Hz, 1.5 hr), 1.16 (t, *J* = 7.6 Hz, 1.5 hr), 1.87 (q, *J* = 8.0 Hz, 1 hr), 2.43 (q, *J* = 7.6 Hz, 1 hr), 3.61–3.78 (m, 2 hr), 3.88–3.91 (m, 1 hr), 6.21 (d, *J* = 3.1 Hz, 0.5), 6.25 (d, *J* = 3.3 Hz, 0.5 hr), 6.48 (d, *J* = 3.3 Hz, 0.5 hr), 6.52 (d, *J* = 7.5 Hz, 0.5 hr) 6.64 (d, *J* = 7.5 Hz, 0.5 hr), 6.98–7.11 (m, 3 hr), 7.24–7.27 (m, 1 hr), 7.33–7.39 (m, 1.5 hr), 7.48–7.57 (m, 2 hr), 8.85 (s, 0.5 hr), 9.01 (s, 0.5 hr). **HRMS:** calc. for [M+H]$^+$ C$_{23}$H$_{20}$N$_2$O$_4$F: 407.14016, found 407.14009

**Chemical structure 4.** (+/-)1-(2-Ethylphenyl)−5-((5-(4-fluorophenyl)furan-2-yl)methyl)pyrimidine-2,4,6 (1H,3H,5H)-trione (racemic mixture) (343139).

## (E/Z)−5-((5-(4-Acetylphenyl)furan-2-yl)methylene)−1-(2-ethylphenyl) pyrimidine-2,4,6 (1H,3H,5H)-trione (1:1 mixture of E/Z isomers) (343870)

Compound **343870** (77%, orange solid) was synthesized as described for **cp028**. **$^1$H NMR (CDCl$_3$, 400MHz):** δ 1.20 (t, J = 7.6 Hz, 3 hr), 2.48–2.54 (m, 2 hr), 2.65 (s, 3 hr), 7.09 (d, J = 4.1 Hz, 0.5 hr), 7.16–7.19 (m, 1.5 hr), 7.34–7.38 (m, 1 hr), 7.41–7.48 (m, 2 hr), 7.94 (d, J = 13.0, 8.4 Hz, 2 hr), 8.06 (dd, J = 8.5, 1.9 Hz, 2 hr), 8.18 (s, 0.5 hr), 8.25 (s, 0.5 hr), 8.53 (s, 0.5 hr), 8.59 (s, 0.5 hr), 8.76 (d, J = 4.1 Hz, 0.5 hr), 8.81 (d, J = 4.1 Hz, 0.5 hr). **$^{13}$C NMR (CDCl$_3$, 100MHz):** δ 13.8, 13.9, 23.9, 26.7, 110.4, 110.6, 112.9, 125.7, 127.1,127.3, 128.7, 128.8, 129.2, 129.2, 129.3, 129.8, 129.9, 131.6, 132.1, 132.3, 132.4, 132.5, 132.9, 137.9, 137.9, 140.5, 140.9, 141.5, 141.6, 149.3, 149.3, 149.4, 151.3, 151.4, 160.3, 160.7, 160.8, 161.5, 161.8, 163.0, 197.1, 197.1 **HRMS:** calc. for [M+H]$^+$ C$_{25}$H$_{21}$N$_2$O$_5$: 429.14450, found 429.14433.

**Chemical structure 5.** (E/Z)−5-((5-(4-Acetylphenyl)furan-2-yl)methylene)−1-(2-ethylphenyl)pyrimidine-2,4,6 (1H,3H,5H)-trione (1:1 mixture of E/Z isomers) (343870).

## (E/Z)−5-((5-(4-(Dimethylamino)phenyl)furan-2-yl)methylene)−1-(2-ethylphenyl) pyrimidine-2,4,6 (1H,3H,5H)-trione (1.5:1 mixture of E/Z isomers) (343869)

Compound **343869** (83%, dark purple solid) was synthesized as described for cp028. **$^1$H NMR (CDCl$_3$, 400MHz):** δ 1.20 (t, J = 7.6 Hz, 3 hr), 2.52 (m, 2 hr), 3.09 (s, 6 hr), 6.79 (dd, J = 9.0, 3.1 Hz, 2 hr), 6.84 (d, J = 4.3 Hz, 0.6 hr), 6.92 (d, J = 4.3 Hz, 0.4 hr), 7.17 (d, J = 8.0 Hz, 0.4 hr), 7.19 (d, J = 7.9 Hz, 0.6 hr), 7.33–7.46 (m, 3 hr), 7.75 (t, J = 8.3 Hz, 2 hr), 7.88 (s, 0.4 hr), 7.95 (s, 0.6 hr), 8.44 (s, 0.4), 8.49 (s, 0.6 hr), 8.85 (d, J = 3.7 Hz, 0.6 hr), 8.90 (bs, 0.4 hr). **HRMS:** calc. for [M+H]$^+$ C$_{25}$H$_{23}$N$_3$O$_4$: 430.17613, found 430.17545.

**Chemical structure 6.** (E/Z)−5-((5-(4-(Dimethylamino)phenyl)furan-2-yl)methylene)−1-(2-ethylphenyl) pyrimidine-2,4,6 (1H,3H,5H)-trione (1.5:1 mixture of E/Z isomers) (343869).

## (E/Z)−5-((5-(3,4-dimethoxyphenyl)furan-2-yl)methylene)−1-(2-ethylphenyl)pyrimidine-2,4,6 (1H,3H,5H)-trione (1:1 mixture of E/Z isomers) (343877)

Compound **343877** (64%, dark red solid) was synthesized as described for **cp028**. $^1$H NMR (CDCl$_3$, **400MHz):** δ 1.20 (t, J = 7.6 Hz, 3 hr), 2.48–2.55 (m, 2 hr), 3.87 (s, 6 hr), 6.55 (t, J = 2.2 Hz, 1 hr), 6.96 (d, J = 4.1 Hz, 0.5 hr), 6.99 (dd, J = 3.7, 2.3 Hz, 2 hr), 7.04 (d, J = 4.0 Hz, 0.5 hr), 7.18 (t, J = 8.6 Hz, 1 hr), 7.35 (ddd, J = 14.7, 2.3 Hz, 1 hr), 7.40–7.48 (m, 2 hr), 8.30 (s, 0.5 hr), 8.39 (s, 0.5 hr), 8.52 (s, 0.5 hr), 8.58 (s, 0.5 hr), 8.76 (d, J = 3.9 Hz, 0.5 hr), 8.83 (d, J = 4.0 Hz, 0.5 hr). **HRMS:** calc. for [M+H]$^+$ C$_{25}$H$_{23}$N$_2$O$_6$: 447.15506, found 447.15477.

**Chemical structure 7.** (E/Z)−5-((5-(3,4-dimethoxyphenyl)furan-2-yl)methylene)−1-(2-ethylphenyl)pyrimidine-2,4,6 (1H,3H,5H)-trione (1:1 mixture of E/Z isomers) (343877).

## (E/Z)−5-((5-(3,5-Dichlorophenyl)furan-2-yl)methylene)−1-(2-ethylphenyl) pyrimidine-2,4,6 (1H,3H,5H)-trione (1:1 mixture of E/Z isomers) (343873)

Compound **343873** (78%, orange solid) was synthesized as described for **cp028**. $^1$H NMR (CDCl$_3$, **400MHz):** δ 1.20 (t, J = 7.6 Hz, 3 hr), 2.48–2.54 (m, 2 hr), 7.00 (d, J = 4.1 Hz, 0.5 hr), 7.07 (d, J = 4.0 Hz, 0.5 hr), 7.17, (t, J = 7.16 Hz, 1 hr), 7.35 (dd, J = 14.3, 6.7 Hz, 1 hr), 7.40–7.38 (m, 3 hr), 7.70 (dt, J = 4.1, 2.1 Hz, 2 hr), 8.23 (s, 0.5 hr), 8.30 (s, 0.5 hr), 8.48 (s, 0.5 hr), 8.55 (s, 0.5 hr), 8.72 (d, J = 4.0 Hz, 0.5 hr), 8.78 (d, J = 4.1 Hz, 0.5 hr). **HRMS:** calc. for [M+H]$^+$ C$_{23}$H$_{17}$Cl$_2$N$_2$O$_4$: 455.05599, found 455.0599. **HRMS:** calc. for [M+H]$^+$ C$_{23}$H$_{17}$Cl$^{37}$ClN$_2$O$_4$: 457.05284, found 457.05304.

**Chemical structure 8.** (E/Z)−5-((5-(3,5-Dichlorophenyl)furan-2-yl)methylene)−1-(2-ethylphenyl) pyrimidine-2,4,6 (1H,3H,5H)-trione (1:1 mixture of E/Z isomers) (343873).

## (E/Z)−1-(2-Ethylphenyl)−5-((5-(3-fluorophenyl)furan-2-yl)methylene) pyrimidine-2,4,6 (1H,3H,5H)-trione (1.2:1 mixture of E/Z isomers) (343867)

Compound **343867** (80%, orange solid) was synthesized as described for **cp028**. $^1$**H NMR (CDCl$_3$, 400MHz):** δ 1.20 (t, $J$ = 7.60 Hz, 3 hr), 2.48–2.55 (m, 2 hr), 6.99 (dd, $J$ = 4.1, 0.7 Hz, 0.55 hr), 7.07 (dd, $J$ = 4.0, 0.7 Hz, 0.45 hr), 7.17 (t, $J$ = 7.9 Hz, 1 hr), 7.32–7.48 (m, 5 hr), 7.48–7.70 (m, 1 hr), 7.81–7.83 (m, 1 hr), 8.50 (s, 0.45 hr), 8.57 (s, 0.55 hr), 8.61 (bs, 0.45 hr), 8.63 (bs, 0.55 hr), 8.75 (d, $J$ = 4.1 Hz, 0.55 hr), 8.82 (d, $J$ = 4.0 Hz, 0.45 hr). **HRMS:** calc. for [M+H]$^+$ C$_{23}$H$_{18}$N$_2$O$_4$F: 405.12451, found 405.12435

**Chemical structure 9.** (E/Z)−1-(2-Ethylphenyl)−5-((5-(3-fluorophenyl)furan-2-yl)methylene)pyrimidine-2,4,6 (1H,3H,5H)-trione (1.2:1 mixture of E/Z isomers) (343867).

## (E/Z)−5-((5-(3-Chlorophenyl)furan-2-yl)methylene)−1-(2-ethylphenyl) pyrimidine-2,4,6 (1H,3H,5H)-trione (1:1 mixture of E/Z isomers) (343871)

Compound **343871** (82%, orange solid) was synthesized as described for **cp028**. $^1$**H NMR (CDCl$_3$, 400MHz):** δ 1.20 (td, $J$ = 7.6, 0.5 Hz, 3 hr), 2.48–2.55 (m, 2 hr), 6.99 (dd, $J$ = 4.1, 0.8 Hz, 0.5 hr), 7.07 (dd, $J$ = 4.1, 0.8 Hz, 0.5 hr), 7.12–7.20 (m, 2 hr), 7.36 (ddd, $J$ = 14.1, 7.6, 2.4 Hz, 1 hr), 7.40–7.49 (m, 3 hr), 7.51–7.56 (m, 1 hr), 7.63 (t, $J$ = 8.2 Hz, 1 hr), 8.01 (s, 0.5 hr), 8.08 (s, 0.5 hr), 8.50 (s, 0.5 hr), 8.56 (s, 0.5 hr), 8.75 (d, $J$ = 4.1 Hz, 0.5 hr), 8.81 (d, $J$ = 4.1 Hz, 0.5 hr). **HRMS:** calc. for [M+H]$^+$ C$_{23}$H$_{17}$ClN$_2$O$_4$: 421.09496, found 421.09481. **HRMS:** calc. for [M+H]$^+$ C$_{23}$H$_{18}$$^{37}$ClN$_2$O$_4$: 423.09201, found 423.09191.

**Chemical structure 10.** (E/Z)−5-((5-(3-Chlorophenyl)furan-2-yl)methylene)−1-(2-ethylphenyl)pyrimidine-2,4,6 (1H,3H,5H)-trione (1:1 mixture of E/Z isomers) (343871).

## (E/Z)−5-(4-Chlorobenzylidene)−1-(2-ethylphenyl)pyrimidine-2,4,6 (1H,3H,5H)-trione (1:1 mixture of E/Z isomers) (1:1 mixture of E/Z isomers) (343878)

To a solution of barbituric acid **C** (0.050 g, 0.213 mmol) in EtOH (2.7 mL, 0.08 M), 4-fluorobenzalde-hye (0.030 g, 0.213 mmol) was added. The mixture was heated at reflux for 2 hr and the solvents were evaporated. The crude product was purified by flash chromatography (1% MeOH/DCM) to obtain product **343878** as a white solid (66.5 mg, 88%). The product was isolated as a 1:1 mixture of the E and Z isomers. $^1$**H NMR (CDCl$_3$, 400MHz):** δ 1.20 (td, $J$ = 7.6, 4.4 Hz, 3 hr), 2.50 (qd, $J$ = 7.6, 2.5 Hz, 2 hr), 7.15 (d, $J$ = 7.9 Hz, 1 hr), 7.35 (t, $J$ = 7.4 Hz, 1 hr), 7.39–7.49 (m, 4 hr), 7.97 (s, 0.5 hr), 8.09 (s, 0.5 hr), 8.15 (t, $J$ = 8.2 Hz, 2 hr), 8.56 (s, 0.5 hr), 8.60 (s, 0.5 hr). **HRMS:** calc. for

[M+H]$^+$ C$_{19}$H$_{16}$ClN$_2$O$_3$: 355.08440, found 355.06739. **HRMS:** calc. for [M+H]$^+$ C$_{19}$H$_{16}$$^{37}$ClN$_2$O$_3$: 357.08145, found 357.07858.

**Chemical structure 11.** (E/Z)−5-(4-Chlorobenzylidene)−1-(2-ethylphenyl)pyrimidine-2,4,6 (1H,3H,5H)-trione (1:1 mixture of E/Z isomers) (1:1 mixture of E/Z isomers) (343878).

## (E/Z)−5-(4-Bromobenzylidene)−1-(2-ethylphenyl)pyrimidine-2,4,6 (1H,3H,5H)-trione (1.7:1 mixture of isomers E/Z) (343879)

Compound **343879** (67%, white solid) was synthesized as described for **343878**. $^1$H NMR (CDCl$_3$, 400MHz): δ 1.13 (td, $J$ = 7.6, 4.8 Hz, 3 hr), 2.43 (qd, $J$ = 7.6, 3.2 Hz, 2 hr), 7.08 (d, $J$ = 8.0 Hz, 1 hr), 7.25–7.30 (m, 1 hr), 7.34–7.40 (m, 1 hr), 7.50 (d, $J$ = 8.6 Hz, 0.63 hr) 7.57 (d, $J$ = 8.6 Hz, 0.37 hr), 7.90 (bs, 0.63 hr), 7.98 (t, $J$ = 8.5 Hz, 2 hr), 8.02 (bs, 0.37 hr), 8.47(s, 0.63 hr), 8.51 (s, 0.37 hr). **HRMS:** calc. for [M+H]$^+$ C$_{19}$H$_{16}$BrN$_2$O$_3$: 399.03388, found 399.03388; for C$_{19}$H$_{16}$$^{81}$BrN$_2$O$_3$: 401.03183, found 401.03182

**Chemical structure 12.** (E/Z)−5-(4-Bromobenzylidene)−1-(2-ethylphenyl)pyrimidine-2,4,6 (1H,3H,5H)-trione (1.7:1 mixture of isomers E/Z) (343879).

## (E/Z)−5-((5-(3,4-Dichlorophenyl)furan-2-yl)methylene)−1-(2-ethylphenyl) pyrimidine-2,4,6 (1H,3H,5H)-trione (1.4:1 mixture of isomers E/Z) (343872)

Compound **343872** (85%, yellow solid) was synthesized as described for **cp028**. $^1$H NMR (CDCl$_3$, 400MHz): δ 1.20 (t, $J$ = 7.6 Hz, 3 hr), 2.48–2.50 (m, 2 hr), 6.97 (d, $J$ = 4.1 Hz, 0.4 hr), 7.04 (d, $J$ = 4.0 Hz, 0.6 hr), 7.17 (t, $J$ = 7.4 Hz, 1 hr), 7.35 (ddd, $J$ = 14.5, 6.9, 2.2 Hz, 1 hr), 7.40–7.48 (m, 2 hr), 7.52 (s, 0.4 hr), 7.54 (s, 0.6 hr), 7.64 (ddd, $J$ = 8.0, 5.8, 2.0 Hz, 1 hr), 7.89 (d, $J$ = 4.1 Hz, 1 hr), 8.47 (s, 0.6 hr), 8.54 (s, 0.4 hr), 8.72 (d, $J$ = 4.1 Hz, 0.4 hr), 8.79 (d, $J$ = 4.0 Hz, 0.6 hr), 9.07 (s, 0.6 hr), 9.13 (s, 0.4 hr). **HRMS:** calc. for [M+H]$^+$ C$_{23}$H$_{17}$Cl$_2$N$_2$O$_4$: 455.05599, found 455.0599. **HRMS:** calc. for [M+H]$^+$ C$_{23}$H$_{16}$Cl$^{37}$ClN$_2$O$_4$: 457.05284, found 457.05304.

**Chemical structure 13.** (E/Z)−5-((5-(3,4-Dichlorophenyl)furan-2-yl)methylene)−1-(2-ethylphenyl) pyrimidine-2,4,6 (1H,3H,5H)-trione (1.4:1 mixture of isomers E/Z) (343872).

## (E/Z)−5-((5-([1,1′-Biphenyl]−4-yl)furan-2-yl)methylene)−1-(2-ethylphenyl)pyrimidine-2,4,6 (1H,3H,5H)-trione (1:1 mixture of isomers E/Z) (343868)

Compound **343868** (85%, orange solid) was synthesized as described for **cp028**. **$^1$H NMR (CDCl$_3$, 400MHz):** δ 1.21 (td, J = 7.6, 1.2 Hz, 3 hr), 2.49–2.56 (m, 2 hr), 7.03 (dd, J = 4.1, 0.8 Hz, 0.5 hr), 7.12 (dd, J = 4.1, 0.8 Hz, 0.5 hr), 7.16–7.21 (m, 1 hr), 7.33–7.50 (m, 6 hr), 7.65 (d, J = 7.1 Hz, 2 hr), 7.73 (d, J = 7.9 Hz, 2 hr), 7.91–7.95 (m, 2 hr), 8.03 (bs, 0.5 hr), 8.09 (bs, 0.5 hr), 8.53 (s, 0.5 hr), 8.59 (s, 0.5 hr), 8.81 (d, J = 4.1 Hz, 0.5 hr), 8.86 (d, J = 3.8 Hz, 0.5 hr). **HRMS:** calc. for [M+H]$^+$ C$_{29}$H$_{23}$N$_2$O$_4$: 463.16523, found 463.16491

**Chemical structure 14.** (E/Z)−5-((5-([1,1′-Biphenyl]−4-yl)furan-2-yl)methylene)−1-(2-ethylphenyl)pyrimidine-2,4,6 (1H,3H,5H)-trione (1:1 mixture of isomers E/Z) (343868).

## 5-((5-(4-fluorophenyl)furan-2-yl)methylene)pyrimidine-2,4,6 (1H,3H,5H)-trione

Compound **343135** (79%, red solid) was synthesised as described for cp028 but using barbituric acid instead of compound **C** (CAS: 67-52-7) **$^1$H NMR (CDCl$_3$, 400MHz):** δ 7.39 (t, J = 8.7 Hz, 2 hr), 7.44 (d, J = 4.1 Hz, 1 hr), 8.03 (dd, J = 8.8, 5.3 Hz, 2 hr), 8.11 (s, 1 hr), 8.55 (d, J = 4.0 Hz, 1 hr), 11.27 (bs, 2 hr). **HRMS:** calc. for [M+H]$^+$ C$_{15}$H$_9$N$_2$O$_4$F: 301.06191, found 301.06206

**Chemical structure 15.** 5-((5-(4-fluorophenyl)furan-2-yl)methylene)pyrimidine-2,4,6 (1H,3H,5H)-trione.

## Acknowledgements

We thank Gabi Heyne, Uwe Plessman, Monika Raabe, Hossein Kohansal and Annika Kuhn for excellent technical assistance. We are grateful to Thomas Conrad for cultivation of HeLa cells, Prakash Dube for help in EM analyses and Kum-Loong Boon for providing anti-MBP antibodies. This work was supported by a grant (LU 294/15–1) from the Deutsche Forschungsgemeinschaft (DFG).

## Additional information

### Funding

| Funder | Grant reference number | Author |
| --- | --- | --- |
| Deutsche Forschungsgemeinschaft | LU 294/15-1 | Reinhard Lührmann |

The funders had no role in study design, data collection and interpretation, or the decision to submit the work for publication.

### Author contributions

AS, Conceptualization, Formal analysis, Validation, Investigation, Writing—original draft, Writing—review and editing; CLW, Conceptualization, Formal analysis, Supervision, Validation, Writing—original draft, Writing—review and editing; MMA, JC, SS, BK, Formal analysis, Validation, Investigation, Writing—review and editing; DEA, TS, Formal analysis, Validation, Investigation; PB, Formal analysis, Investigation; HU, Supervision, Validation, Project administration; HW, Conceptualization, Resources, Supervision, Project administration, Writing—review and editing, ; RL, Conceptualization, Resources, Supervision, Funding acquisition, Visualization, Project administration, Writing—review and editing

### Author ORCIDs

Cindy L Will, http://orcid.org/0000-0003-0012-9106
Sonja Sievers, http://orcid.org/0000-0003-0854-4507
Reinhard Lührmann, http://orcid.org/0000-0003-3253-7518

## Additional files

### Supplementary files

- Supplementary file 1. Protein composition of $A^{028}$ and $B^{028}$ complexes as determined by mass spectrometry. Proteins identified by LC-MS/MS in human spliceosomal B and $B^{act}$ complexes, as well as complexes stalled in the presence of compound 028 ($B^{028}$). Total spectral counts of sequenced peptides are shown. Peptides and proteins were identified by searching fragment spectra against the NCBI database (taxonomy human) using Mascot as search engine and were annotated with Scaffold software. Proteins are grouped according to function or association. Common contaminants, such as ribosomal proteins, are not shown.

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
