## [Decision Letter]

Thank you for submitting your article "Identification of a small molecule inhibitor that stalls splicing at an early step of spliceosome activation" for consideration by *eLife*. Your article has been favorably evaluated by James Manley (Senior Editor) and three reviewers, one of whom, Timothy W Nilsen (Reviewer #1), is a member of our Board of Reviewing Editors.

The reviewers have discussed the reviews with one another and the Reviewing Editor has drafted this decision to help you prepare a revised submission.

The reviewers were generally quite positive about the work which identifies a novel intermediate in the progression from B complex (inactive) to the B^act^ spliceosomal complex (activated) using a new chemical inhibitor of splicing. Despite the enthusiasm for the studies, the following revisions were deemed to be necessary.

1) The inhibitory action of cp028 on an unrelated mammalian splicing substrate should be demonstrated.

2) The conclusion at the end of the subsection “Compound 028 stalls spliceosome assembly at a stage between B and B^act^” should be softened to include the possibility that cp028 might promote the assembly of an atypical complex that nonetheless can be chased by incubation with fresh MN-treated extract.

3) Results of chase experiments wherein MN-treated extracts either plus or minus cp028 should be shown.

4) A much more thorough description of the chemical screen must be included. A previous paper referenced does not really described how the screen worked… What was their "automated set-up"? What are all the details of the reactions (wells, volumes, reaction times etc.)? What libraries were screened? What kinds of compounds did they contain and why were they chosen? What statistical tests were applied to define positive hits? What kinds of false positives were generated? Were there many that did not pass secondary validation? What secondary validations were done besides the in vitro splicing reactions shown? This appears to be one of the more successful screens for splicing inhibitors, and it needs to be compared to other such studies, as well as described in sufficient detail to reproduce. Similarly, the authors need to report the necessary information on each commercially available compound to obtain it without having to do structure-based searches. The compounds are reported to be from Chemdiv but the numbers given do not yield hits in a search of their catalog.

---

## [Author Response]

*The reviewers were generally quite positive about the work which identifies a novel intermediate in the progression from B complex (inactive) to the B^act^ spliceosomal complex (activated) using a new chemical inhibitor of splicing. Despite the enthusiasm for the studies, the following revisions were deemed to be necessary.*

*1) The inhibitory action of cp028 on an unrelated mammalian splicing substrate should be demonstrated.*

We performed in vitro splicing assays with a second, unrelated substrate, an IgM pre-mRNA (Guth et al., 1999) – which revealed a similar inhibitory effect of cp028 on splicing and splicing complex formation. These data are now shown in Figure 1—figure supplement 1, panel C and are mentioned in the last paragraph of the subsection “Identification of a novel small molecule that inhibits pre-mRNA splicing in vitro”. The original high throughput screen for small molecule inhibitors of splicing that identified cp028 was performed with the PM5 pre- mRNA – a truncated version of the PYP pre-mRNA (Wollerton et al., 2004, Mol Cell) – which is also not related to the MINX or IgM pre-mRNA substrates. Thus, cp028 inhibits the in vitrosplicing of at least 3 distinct pre-mRNA substrates.

*2) The conclusion at the end of the subsection “Compound 028 stalls spliceosome assembly at a stage between B and B^act^” should be softened to include the possibility that cp028 might promote the assembly of an atypical complex that nonetheless can be chased by incubation with fresh MN-treated extract.*

We softened our conclusion at the end of the subsection “Compound 028 stalls spliceosome assembly at a stage between B and B^act^” and now mention in the Discussion subsection “A new tool to dissect pre-mRNA splicing at the spliceosome activation stage”, that we cannot rule out that cp028 promotes the assembly of an atypical complex that nonetheless can be chased by incubation with fresh MN-treated extract.

*3) Results of chase experiments wherein MN-treated extracts either plus or minus cp028 should be shown.*

We have now performed chase experiments where we additionally include cp028 in the MN- treated extract. Whereas the B and B^028^ complexes can be chased into catalytically-active spliceosomes with MN-treated extract lacking cp028, they cannot be chased when cp028 is added to the MN-treated extract. This indicates that cp028 binds to/inactivates a splicing factor required for the progression of the B/B^028^ complexes to catalytically active spliceosomes. We mention this now in the Results subsection “B^028^ complexes can be chased into catalytically-active spliceosomes”. We have also replaced the original Figure 4, with a new version showing additionally the chase with MN-treated extract plus cp028.

*4) A much more thorough description of the chemical screen must be included. A previous paper referenced does not really described how the screen worked… What was their "automated set-up"? What are all the details of the reactions (wells, volumes, reaction times etc.)? What libraries were screened? What kinds of compounds did they contain and why were they chosen? What statistical tests were applied to define positive hits? What kinds of false positives were generated? Were there many that did not pass secondary validation? What secondary validations were done besides the* in vitro *splicing reactions shown? This appears to be one of the more successful screens for splicing inhibitors, and it needs to be compared to other such studies, as well as described in sufficient detail to reproduce. Similarly, the authors need to report the necessary information on each commercially available compound to obtain it without having to do structure-based searches. The compounds are reported to be from Chemdiv but the numbers given do not yield hits in a search of their catalog.*

We now include in the Materials and methods a detailed description of the high throughput screen for small molecule inhibitors of pre-mRNA splicing and also provide the catalogue numbers for all of the compounds that we purchased from ChemDiv.